# PRMT5 triggers glucocorticoid-induced cell migration in triple-negative breast cancer

Lara Malik Noureddine[1,2,3,4], Julien Ablain[1,2,3], Ausra Surmieliova-Garnès[1,2,3], Julien Jacquemetton[1,2,3], Thuy Ha Pham[1,2,3], Elisabetta Marangoni[5], Anne Schnitzler[6], Ivan Bieche[6], Bassam Badran[4], Olivier Trédan[1,2,3,7], Nader Hussein[4], Muriel Le Romancer[1,2,3,*] ⓘ, Coralie Poulard[1,2,3,*] ⓘ

**Triple-negative breast cancers (TNBCs) are the most aggressive breast cancers, and therapeutic options mainly rely on chemotherapy and immunotherapy. Although synthetic glucocorticoids (GCs) are given to alleviate the side effects of these treatments, GCs and their receptor, the glucocorticoid receptor (GR), were recently associated with detrimental effects, albeit the mechanisms involved remain elusive. Here, we identified the arginine methyltransferase PRMT5 as a master coregulator of GR, serving as a scaffold protein to recruit phospho-HP1γ and subsequently RNA polymerase II, independently of its methyltransferase activity. Moreover, the GR/PRMT5/HP1γ complex regulated the transcription of GC-target genes involved in cell motility and triggering cell migration of human TNBC cells in vitro and in a zebrafish model. Of note, we observed that GR/PRMT5 interaction was low in primary tumors but significantly increased in residual tumors treated with chemotherapy and GCs in neoadjuvant setting. These data suggest that the routine premedication prescription of GCs for early TNBC patients should be further assessed and that this complex could potentially be modulated to specifically target deleterious GR effects.**

## Introduction

Breast cancer (BC) is a leading cause of death in women worldwide. In 2020, 2.3 million women were diagnosed with BC and around 685,000 deaths were recorded according to the International Agency for Research on Cancer (Sung et al, 2021). BCs are routinely classified in the clinic according to the expression of three markers: estrogen receptor $\alpha$ (ER$\alpha$), progesterone receptor (PR), and human EGF receptor HER2. Triple-negative breast cancers (TNBCs) (ER$\alpha$-/PR-/HER2-) represent around 15% of BCs, and chemotherapy is the cornerstone of the treatment in the (neo)-adjuvant setting. Because the results of the phase III KEYNOTE-522 trial, pembrolizumab, targeting PD-1 (programmed death-1), in combination with standard carboplatin/paclitaxel followed by anthracycline-based chemotherapy was approved for stage II/III BC patients (Schmid et al, 2022). Chemo-immunotherapy combinations have thus become a standard-of-care of early TNBC, regardless of PD-1 tumor expression.

Synthetic glucocorticoids (GCs) play an important role in alleviating patients with side effects of cytotoxic chemotherapies, including nausea, fatigue, and/or allergic events. However, the use of prednisone or dexamethasone (Dex) has a detrimental impact on patient outcome according to several studies on different cancer types treated with immunotherapy-based treatments, likely because of the fact that GCs decrease the number of cytotoxic lymphocytes (Maslov et al, 2021). The routine implementation of GC premedication thus raises concern. Furthermore, GCs have recently been associated with tumorigenic effects, such as the development of metastases and resistance to chemotherapy (Chen et al, 2015; Obradović et al, 2019; Noureddine et al, 2021), although the causality between GCs and these adverse effects has not been fully demonstrated and the mechanisms involved are still poorly understood.

The effects of GCs are mediated by the binding of the glucocorticoid receptor (GR) to a specific set of coregulatory proteins for a given function. GR expression has different prognostic values depending on the BC subtype; a high GR expression is correlated with a better prognosis in early-stage ER$\alpha$+ BC but with a worse prognosis in TNBC (Pan et al, 2011; West et al, 2016). Likewise, at the transcriptional level, GCs inhibit ER$\alpha$ transcriptional activity and E$_2$-mediated cell proliferation in ER$\alpha$+ BC (Karmakar et al, 2013; West et al, 2016; Yang et al, 2017), but drive the expression of pro-tumorigenic genes in TNBC (Chen et al, 2015). GR and ER$\alpha$ are hormone-regulated transcription factors that regulate transcription by recruiting coregulator proteins to the promoter/enhancer

---

[1]Université de Lyon, Lyon, France  [2]Inserm U1052, Centre de Recherche en Cancérologie de Lyon, Lyon, France  [3]CNRS UMR5286, Centre de Recherche en Cancérologie de Lyon, Lyon, France  [4]Lebanese University, Faculty of Sciences I, Department of Chemistry and Biochemistry, Laboratory of Cancer Biology and Molecular Immunology, Beirut, Lebanon  [5]Institut Curie, Translational Research Department, PSL University, Paris, France  [6]Institut Curie, Department of Genetics, Paris, France  [7]Centre Leon Bérard, Oncology Department, Lyon, France

Correspondence: coralie.poulard@lyon.unicancer.fr; muriel.leromancer@lyon.unicancer.fr
*Muriel Le Romancer and Coralie Poulard contributed equally to this work

---

regions of their target genes. Coregulators participate in remodeling the chromatin structure and in promoting or inhibiting the recruitment and activation of RNA polymerase II. Most of the known coregulators were discovered either for their role in transcriptional activation or repression. However, many coregulators contribute to both functions, depending on the specific gene targeted and cellular environment (Stallcup & Poulard, 2020).

Recent studies on GR and other transcription factors demonstrated that specific coregulators are preferentially required for genes involved in selected physiological responses among multiple pathways that are regulated by a given transcription factor (Wu et al, 2014; Stallcup & Poulard, 2020). The three homologous members of the p160 coregulator family (SRC-1, SRC-2, and SRC-3) represent a good example of this concept. Even if they share many target genes, the individual knockout of these three coregulators in mice results in different phenotypes, indicating that each SRC protein regulates distinct physiological pathways (Xu & Li, 2003). For instance, only SRC-2/GRIP1 serves as a corepressor for GR-regulated cytokine genes in macrophages, facilitating the anti-inflammatory effects of GC in vivo (Chinenov et al, 2012). Likewise, HP1γ (CBX3), mainly known for its role in transcriptional repression, was also shown to act as a coactivator (Koike et al, 2000; Lomberk et al, 2006; Kwon et al, 2010; Poulard et al, 2017) after its recruitment through the automethylation of the histone methyltransferases G9a/GLP (EHMT1/2) to regulate migration of lung cancer cells A549 (Poulard et al, 2017), and GC-induced cell death in leukemia (Poulard et al, 2018, 2019).

Modulating the activity of a specific coregulator could thus affect GC regulation of only the subset of GR target genes that require this specific coregulator for a specific physiological pathway. Hence, deciphering the mechanisms that control gene-specific actions of GR coregulators in BC is of utmost importance for the identification of possibly druggable physiological functions. It is now well established that GR may have oncogenic properties in breast tumors and particularly in TNBC. However, directly targeting GR activity is not an option because of its pleiotropic effects in the homeostasis of the organism. For these reasons, we aimed at decrypting the molecular mechanisms associated with the deleterious effects of GR, with a particular focus on coregulators.

Here, we demonstrate that the arginine methyltransferase PRMT5 acts as a key coregulator of GR, independently of its catalytic activity, allowing the recruitment of Phospho-HP1γ and subsequently RNA polymerase II in TNBC. Interestingly, we highlight that the GR/PRMT5/HP1γ complex drives the migratory properties induced by GCs in vitro and in vivo through a specific transcriptional program.

# Results

## HP1γ is a bona fide coactivator of GR in TNBC

To identify potential coregulators of GR implicated in GR signaling in BC, we initially conducted Kaplan-Meier plots of patient relapse-free survival using the Gene Expression Omnibus, EGA, and TCGA databases (Győrffy, 2021). Among the different candidates examined, known as well-described GR coregulators in different systems,

HP1γ was the only coregulator, when combined to GR, to significantly impact TNBC patient survival (Figs 1A and S1A–I). Indeed, although the individual expression of GR or HP1γ was not associated with patient survival (Fig 1B and C), their combined high expression was significantly associated with a shorter relapse-free survival in TNBC patients ($P$ = 0.004) (Fig 1A), suggesting the involvement of HP1γ in GR signaling in BC.

We then searched for GR/HP1γ interactions in different subtypes of TNBC cell lines using proximity ligation assay (PLA), including mesenchymal-like, basal-like, and luminal AR. Upon treatment with Dex, a synthetic GC, GR interacted significantly with HP1γ in the nucleus of all TNBC cell types tested, independently of the level of GR protein within cells (Fig 1D). The specificity of these interactions was validated using an siRNA approach in two cell lines displaying high (MDA-MB-231) and low (HCC-1937) GR levels (Fig S2A and B). Moreover, the addition of other GR agonists (prednisolone and hydrocortisone) led to similar results as Dex (Fig S2C), whereas the addition of the GR antagonist RU486 significantly disrupted these interactions (Fig S2C).

Though HP1γ is mainly described as a corepressor, several reports indicate that it acts as a coactivator when phosphorylated on residue S93 (Koike et al, 2000; Lomberk et al, 2006; Kwon et al, 2010; Poulard et al, 2017). To ascertain whether this function was also activated in GR signaling, we analyzed the interaction between GR and p-S93-HP1γ by PLA and found that GR/p-S93-HP1γ increased upon Dex treatment (Fig 1E). Depletion of either protein by siRNA eliminated most of the signal, validating the specificity of these interactions (Fig 1E). As p-S93-HP1γ was shown to interact with phosphorylated RNA polymerase II (Lomberk et al, 2006), we analyzed the interaction between GR and RNA polymerase II phosphorylated on both S2 and S5 (p-S2/S5-RNA polymerase II) by PLA upon Dex treatment. This interaction (i) strongly increased after treatment, (ii) was abolished upon depletion of GR, and (iii) significantly decreased after HP1γ depletion (Fig 1F). Altogether, these data support that HP1γ acts as a coactivator of GR in TNBC.

## PRMT5 is required for HP1γ recruitment on GR, independently of its enzymatic activity

It was previously demonstrated that HP1γ functions as a coactivator of GR after its recruitment through the histone lysine methyltransferases G9a and GLP (EHMT2 and EHMT1, respectively) in lung adenocarcinoma and B-cell acute lymphoblastic leukemia (Poulard et al, 2017, 2018, 2019). To test this hypothesis in BC, we depleted G9a or GLP in two different TNBC cell lines and studied their impact on GR/HP1γ interactions. Conversely to what was observed in other cancers, in TNBC, G9a and GLP were not essential for these interactions, reinforcing the idea that the involvement of a given coregulator is tissue specific (Fig S3A–D).

As the GST pull-down approach indicated that GR was unable to bind directly to HP1γ using the full-length protein (Fig S4A), we analyzed the interaction between HP1γ and different GR fragments (Fig S4B). We were unable to detect any interaction (Fig S4C). As a positive control, Grip1, a well-known coactivator of GR, was shown to interact with domain 1 of GR (Fig S4D) (Kumar & Thompson, 2003). We then investigated how HP1γ was recruited on GR. Previous work from our team demonstrated that GR could be methylated by the arginine methyltransferase PRMT5 in the nucleus of BC cells

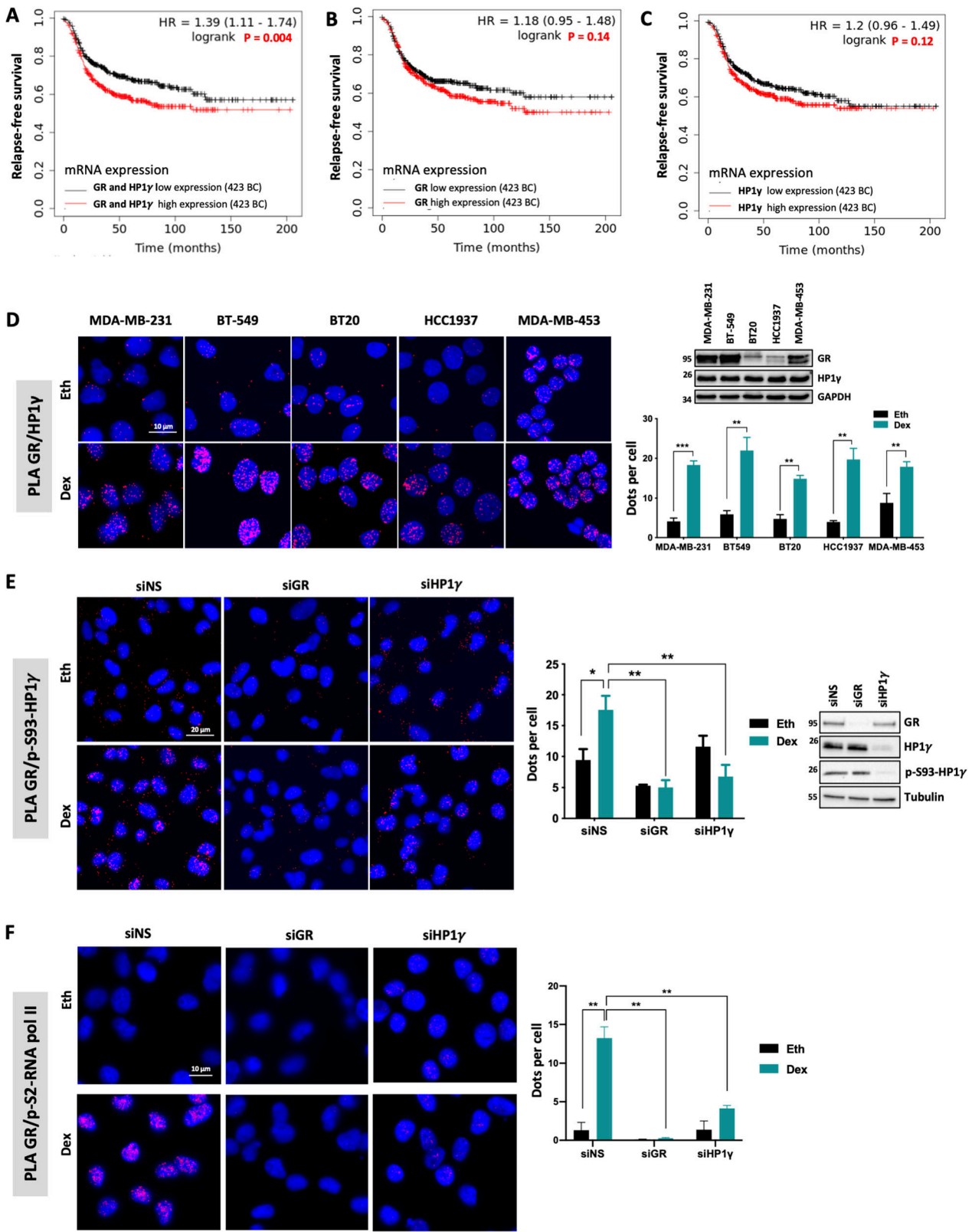

**Figure 1. HP1γ acts as a coactivator of GR in triple-negative breast cancer.**
**(A, B, C)** Relapse-free survival in a cohort of 846 patients with basal breast cancers (BC), with low (black) or high (red) (A) GR and HP1γ expression, (B) GR expression alone, or (C) HP1γ expression alone. Kaplan–Meier analyses conducted on Gene Expression Omnibus, EGA, and TCGA datasets. **(D)** PLA was conducted in different triple-negative breast cancer cell lines to analyze the interaction of endogenous GR and HP1γ. Cells were treated with 100 nM dexamethasone (Dex) or the equivalent volume of

(Poulard et al, 2020), suggesting that PRMT5 could be a coregulator of GR. Interestingly, Fig 2A revealed that GR interacts with PRMT5 in the cytoplasm of MDA-MB-231 cells without ligand and translocates to the nucleus upon hormone treatment. We also confirmed via a siRNA approach that PRMT5 interacts specifically with HP1γ after Dex treatment (Fig 2B). To ascertain that GR forms a tripartite complex with PRMT5 and HP1γ, we used both co-immunoprecipitation (CoIP) and GST pull-down experiments. After overexpressing PRMT5, GR and HP1γ in cells, we immunoprecipitated PRMT5 and found both GR and HP1γ (Fig 2C). Similarly, after immunoprecipitating HP1γ, we found GR and PRMT5 and after immunoprecipitating GR, we found PRMT5 and HP1γ (Fig 2C). Using semi-endogenous CoIP, we found that HP1γ interacted both with GR and PRMT5 (Fig S4E). In addition, because the catalytic activity of PRMT5 is dependent on MEP50, we analyzed whether MEP50 was part of the complex and found that MEP50 interacted both with GR, HP1γ, and PRMT5 (Fig 2C).

Likewise, GST pull-down experiments demonstrated that PRMT5 interacted directly with GR (Fig 2D) and HP1γ (Fig 2E). Based on the GR domain approach, we were able to demonstrate that PRMT5 interacts with the GR1 domain (Fig S4F), as the coactivator Grip1 (Fig S4D), a domain containing the AF-1 region, well-known to bind coregulators, chromatin modulators, and basal transcription machinery (Kumar & Thompson, 2003). Finally, we wondered whether the ternary complex is ligand-dependent. We treated MDA-MB-231 cells with Dex and following CoIP experiments, we found that PRMT5 interacted with (i) HP1γ upon Dex treatment and (ii) with GR independently of Dex treatment (Fig 2F). These results are in accordance with previous results presented in the study as we saw that (i) PRMT5/HP1γ interaction increases after Dex treatment by PLA (Fig 2B) and (ii) that the localization of the interaction GR/PRMT5 changes upon Dex treatment in PLA (Fig 2B) and is not affected by Dex treatment in GST pull-down (Fig 2D). These data indicate that PRMT5 and HP1γ interact with GR in MDA-MB-231 cells upon Dex treatment.

Having shown that HP1γ does not interact directly with GR (Fig S4A and C), we investigated whether PRMT5 could mediate this interaction, by depleting PRMT5 in MDA-MB-231 and HCC-1937 cells (Fig 3A and B). Interestingly, we observed a strong decrease in GR/HP1γ interaction, indicating that PRMT5 is required for the interaction between these two proteins. We also validated that PRMT5 depletion did not impact GR or HP1γ localization (Fig S5A and B). As expected, GR was mainly localized in the cytoplasm without ligand and translocated to the nucleus upon Dex treatment. HP1γ remains a nuclear protein. PRMT5 was mainly localized in the cytoplasm, with a small pool of protein in the nucleus. In addition, PRMT5 depletion did not affect GR or HP1γ localization (Fig S5A and B). As

previously demonstrated for HP1γ, PRMT5 was also essential for the interaction between GR and p-S2/S5-RNA polymerase II following Dex treatment (Fig 3C). As (i) methylation often constitutes a platform for protein recruitment and (ii) PRMT5 was shown to methylate GR (Poulard et al, 2020), we then tested whether this event is involved in the formation of this complex. Surprisingly, our results demonstrated that the catalytic activity of PRMT5 was not involved in HP1γ recruitment with GR (Fig 3D), as a specific PRMT5 inhibitor called GSK595, inhibited the general symmetric dimethylation of the proteins without affecting GR/HP1γ interactions. PRMT5 thus function as a scaffold protein for GR.

## PRMT5 and HP1γ are involved in the migratory function of GC

To characterize the effect of PRMT5 and HP1γ on endogenous target genes that are induced by Dex-activated GR, we performed RNA interference and RNA-sequencing experiments. RNAs were prepared from the MDA-MB-231 cell line expressing siRNA against HP1γ, PRMT5, or a non-specific sequence (siNS) and treated either with 100 nM Dex or vehicle ethanol for 8 h. In three biological replicates, HP1γ and PRMT5 were efficiently depleted by the relevant siRNA (Fig S6A). We identified 275 genes for which mRNA levels changed significantly by at least 1.5-fold, either the levels increased (for 169 genes) or decreased (for 106 genes) in siNS cells after 8 h of Dex treatment (Fig 4A, comparison A, red circle). We then identified a subset of Dex-regulated genes that require HP1γ and PRMT5. As previously described (Poulard et al, 2018), the Dex-induced phenotype was determined by the levels of gene products after Dex treatment more than the Dex-induced fold change. Hence, we analyzed the effect of PRMT5 and HP1γ depletion by comparing gene expression in siNS versus siHP1γ (Fig 4A, comparison B, green circle) or siPRMT5 (Fig 4A, comparison C, blue circle). Differentially expressed genes were defined as those with a significant *P*-value ($P < 0.05$), to maximize the number of genes discovered and gain more statistical power for subsequent analyses. By overlapping the three sets of genes, we identified 89 overlapping genes (Fig 4A, central red area), classified as Dex-regulated, HP1γ/PRMT5-dependent genes (Table S1).

A gene ontology analysis of these 89 genes unveiled an enrichment in genes involved in cell migration and motility (Fig 4B), including specific genes involved in migratory or invasive properties of tumor cells, such as SERPINE1, CCBE1, IGFBP3, or PLAT. Reverse transcriptase followed by quantitative PCR (RT-qPCR) analyses confirmed that depletion of PRMT5 and HP1γ significantly decreased Dex-induced expression levels of SERPINE1, CCBE1, PLAT, and IGFBP3, which were identified as PRMT5- and HP1γ-dependent in the RNA-seq analysis (Fig 4C). Of note, the expression of these

vehicle ethanol (Eth) for 2 h. After cell fixation, PLA with antibodies against GR and HP1γ was performed. The detected interactions are indicated by red dots. The nuclei were counterstained with DAPI (blue). The number of interactions detected by ImageJ analysis is shown as the mean ± SEM of three independent experiments. *P*-value was determined using a paired *t* test. ***$P ≤ 0.001$, **$P ≤ 0.01$. Whole-cell extracts were analyzed for GR, HP1γ, and GAPDH expression by immunoblot. **(E)** PLA was conducted to analyze the interaction of endogenous GR and p-S93-HP1γ after transfection of MDA-MB-231 cells with SMART-pool siRNA targeting GR (siGR), HP1γ, (siHP1γ), or non-specific sequence, and following treatment with 100 nM Dex or the equivalent volume of Eth for 2 h. Detected interactions are shown as the mean ± SEM of three independent experiments. *P*-value was determined using a paired *t* test. *$P ≤ 0.05$, **$P ≤ 0.01$. Whole-cell extracts were analyzed for GR, HP1γ, p-S93-HP1γ, and GAPDH expression by immunoblot. **(F)** PLA was performed to study the interaction of endogenous GR and p-S2/S5-RNApol II after transfection of MDA-MB-231 cells with SMART-pool siRNA targeting GR (siGR), HP1γ (siHP1γ), or non-specific sequence, and following treatment with 100 nM Dex or the equivalent volume of Eth for 2 h. Detected interactions are shown as the mean ± SEM of three independent experiments. *P*-value was determined using a paired *t* test. **$P ≤ 0.01$.

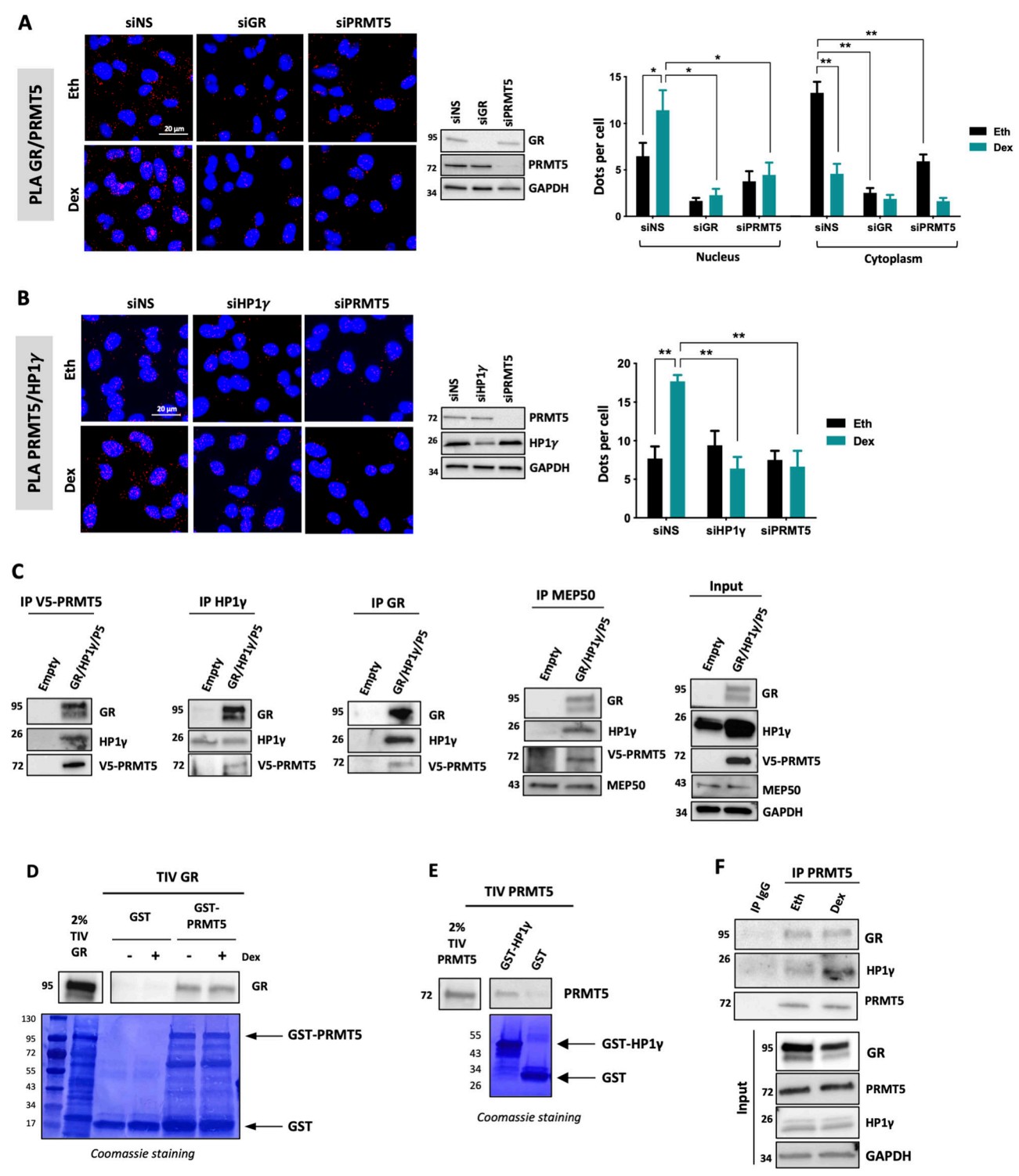

**Figure 2. GR forms a tripartite complex with HP1γ and PRMT5.**
**(A)** To analyze the interaction of endogenous GR and PRMT5 by PLA, MDA-MB-231 cells were transfected with SMART-pool siRNA targeting GR (siGR), PRMT5 (PRMT5), or non-specific sequence and treated with 100 nM dexamethasone (Dex) or the equivalent volume of vehicle ethanol (Eth) for 2 h. After cell fixation, PLA was performed with antibodies against GR and PRMT5. The detected interactions are indicated by red dots. The nuclei were counterstained with DAPI (blue). The number of interactions in the nucleus and cytosol detected by ImageJ analysis is shown as the mean ± SEM of three independent experiments. *P*-value was determined using a paired *t* test. *$P \le 0.05$, **$P \le 0.01$. Whole-cell extracts were analyzed for GR, PRMT5, and GAPDH expression by immunoblot. **(B)** PLA was conducted to analyze endogenous interactions between HP1γ and PRMT5. MDA-MB-231 cells were transfected with SMART-pool siRNA targeting HP1γ (siHP1γ), PRMT5 (PRMT5), or non-specific sequence and treated with 100 nM Dex or the equivalent Eth for 2 h. Detected interactions are shown as the mean ± SEM of three independent experiments. *P*-value was determined using a paired *t* test. **$P \le 0.01$. Whole-cell extracts were analyzed for PRMT5, HP1γ, and GAPDH expression by immunoblot. **(C)** Cos-7 cells were transfected with empty plasmids or plasmids

target genes was also significantly affected by the inhibition of PRMT5 and HP1γ without any Dex induction, likely due to the presence of a pool of PRMT5 and HP1γ on chromatin in basal condition. However, we emphasize the fact that the Dex-induced phenotype is determined by the levels of gene products. In parallel, we confirmed that these target genes were GR-dependent, as the induction of their expression was abolished after GR depletion (Fig S6B). We then investigated whether the catalytic activity of PRMT5 could be involved in the regulation of these target genes (Poulard et al, 2020; Motolani et al, 2021). However, the catalytic activity of PRMT5 was not involved in this process (Fig 4D), suggesting that PRMT5 recruits HP1γ independently of its catalytic activity, likely by acting as a scaffold protein.

Next, to determine if these genes were direct targets of HP1γ and PRMT5 coregulators, we initially validated the Dex-induced binding of GR to neighboring GR response elements identified on published GR ChiP-seq databases using ChIP-qPCR (Fig 5A). Using PRMT5 and HP1γ antibodies, we found by ChIP-qPCR that they were also recruited to the same GR response elements after Dex treatment (Fig 5B and C). PRMT5 recruitment was validated by siRNA approach (Fig S6C), and HP1γ recruitment was previously validated (Poulard et al, 2017). We previously demonstrated that HP1γ and PRMT5 are responsible for the Dex-induced interaction between RNA polymerase II and GR (Figs 1F and 3C). As p-S93-HP1γ was shown to recruit RNA polymerase II, we analyzed the Dex-induced occupancy by RNA polymerase II at the transcription start site of PRMT5/HP1γ-dependent GR target genes by ChIP-qPCR. We found that this recruitment was strongly reduced by depleting HP1γ (Fig 5D). This result indicates that Dex induces binding of the GR/PRMT5/HP1γ complex to chromatin to facilitate the recruitment of RNA polymerase II to the transcription start site for the full transcriptional activation of these genes.

GCs were shown to drive metastasis formation in TNBC, specifically in MDA-MB-231 cells (Obradović et al, 2019). As our results clearly demonstrate that the GR/PRMT5/HP1γ complex regulates a subset of Dex-regulated target genes involved in cell migration, we monitored the cell migratory properties of MDA-MB-231 cells under Dex treatment, following the depletion of our proteins of interest, by X-CELLigence. This method records the cell migratory process in real time without requiring any labeling. When the cells migrate from the upper chamber through the membrane into the bottom chamber, cells create contacts and adhere to the electronic sensors under the membrane, increasing the impedance. As changes in impedance are continuously recorded by the Real-Time Cell Analyzer instrument, cell migration can be monitored in real time via the cell-index profile (Bird & Kirstein, 2009). We first validated Dex-induced migration of MDA-MB-231 cells using this technique and observed that GR depletion abolished this effect, demonstrating that the cell migratory property induced by GCs was driven by GR

(Fig 6A). We then depleted HP1γ or PRMT5 and observed that both depletions significantly decreased cell migration induced by Dex (Fig 6B and C), strongly suggesting that the three proteins (GR, PRMT5, and HP1γ) are involved in cell migration induced by GCs. Furthermore, we confirmed once again that the catalytic activity of PRMT5 was not involved in this process (Fig 6D).

We then challenged our results in vivo, using a zebrafish model (*Danio rerio*) to follow cell migration and invasion in real time in a living vertebrate organism over a few days (Roth et al, 2021). As a major actor of chromatin stability, the stable depletion of HP1γ was not an option for such studies. MDA-MB-231 cells were thus transiently transfected with siRNAs against HP1γ and PRMT5 and treated or not with Dex. The four populations of cells were then labeled using a green dye (DiO) and injected into the yolk sac of 2-d-old zebrafish embryos (Fig 6E). Cell dissemination from the site of injection to the caudal plexus via blood circulation could thus be monitored by fluorescence imaging over a 3-d period (Fig 6F). We quantified the number of metastatic cells in each embryo as a readout of the migratory properties of TNBC cells. Our results demonstrated that Dex increased cell migration and invasion compared with untreated cells (Eth) (Fig 6G). In addition, depletion of HP1γ and PRMT5 significantly decreased the number of metastatic cells upon Dex treatment 3 d post-transplantation (Fig 6G). As indicated in Fig 6E, at the time of transplantation, a pool of labeled cells from each condition was plated to measure protein expression on the day cell migration was assessed (day 5), evidencing sustained down-regulation of HP1γ and PRMT5 throughout the assay (Fig S7). In conclusion, we demonstrated the involvement of GR/PRMT5/HP1γ complex in the migratory properties of TNBC cells under GC treatment.

## The GR/PRMT5 interaction is increased in tumors of patients treated with Dex

We then investigated GR/PRMT5 interactions in 442 patients with invasive BCs sampled at diagnosis including all BC subtypes; 75.9% of luminal A, 13% of luminal B, 4.6% of HER2 enriched, and 6.5% of TNBC. Interactions between GR and PRMT5 in tumors remained very low and no major changes were observed among the different patients (Fig S8A). Next, to circumvent the small number of TNBC samples in a non-selective BC patient cohort, we performed analyses on patient-derived xenografts (PDX) of BC, which show a better engraftment for TNBC samples. Some of them were generated from primary tumors and others from residual tumors after neoadjuvant treatment with chemotherapy complemented with GCs.

In 148 previously characterized PDX models containing luminal B (19%), HER2+ (5%), and TNBC (76%) (Marangoni et al, 2007), we first analyzed GR, PRMT5, and HP1γ expression at the RNA level and found that GR expression remained constant in the three

---

encoding PRMT5, GR, and HP1γ. Lysates were immunoprecipitated (IP) with indicated antibodies and immunoblotted with GR, PRMT5, and HP1γ antibodies. Expression of GR, PRMT5, HP1γ, MEP50, and GAPDH in cell extracts is shown on the right (Input). **(D)** GST and GST-PRMT5 fusion proteins were incubated with in vitro–translated GR, in addition to Dex (100 nM) when indicated and the interaction was then visualized by Western blotting using an anti-GR antibody. The corresponding Coomassie-stained gel is shown in the panel below. **(E)** GST and GST-HP1γ fusion proteins were incubated with in vitro–translated PRMT5, the interaction was then visualized by Western blotting using an anti-PRMT5 antibody. The corresponding Coomassie-stained gel is shown in the panel below. **(F)** MDA-MB-231 cells were incubated with 100 nM of Dex for 8 h. Lysates were immunoprecipitated (IP) with PRMT5 and immunoblotted with GR, PRMT5, and HP1γ antibodies. Expression of GR, PRMT5, HP1γ, and GAPDH in cell extracts is shown underneath (Input).

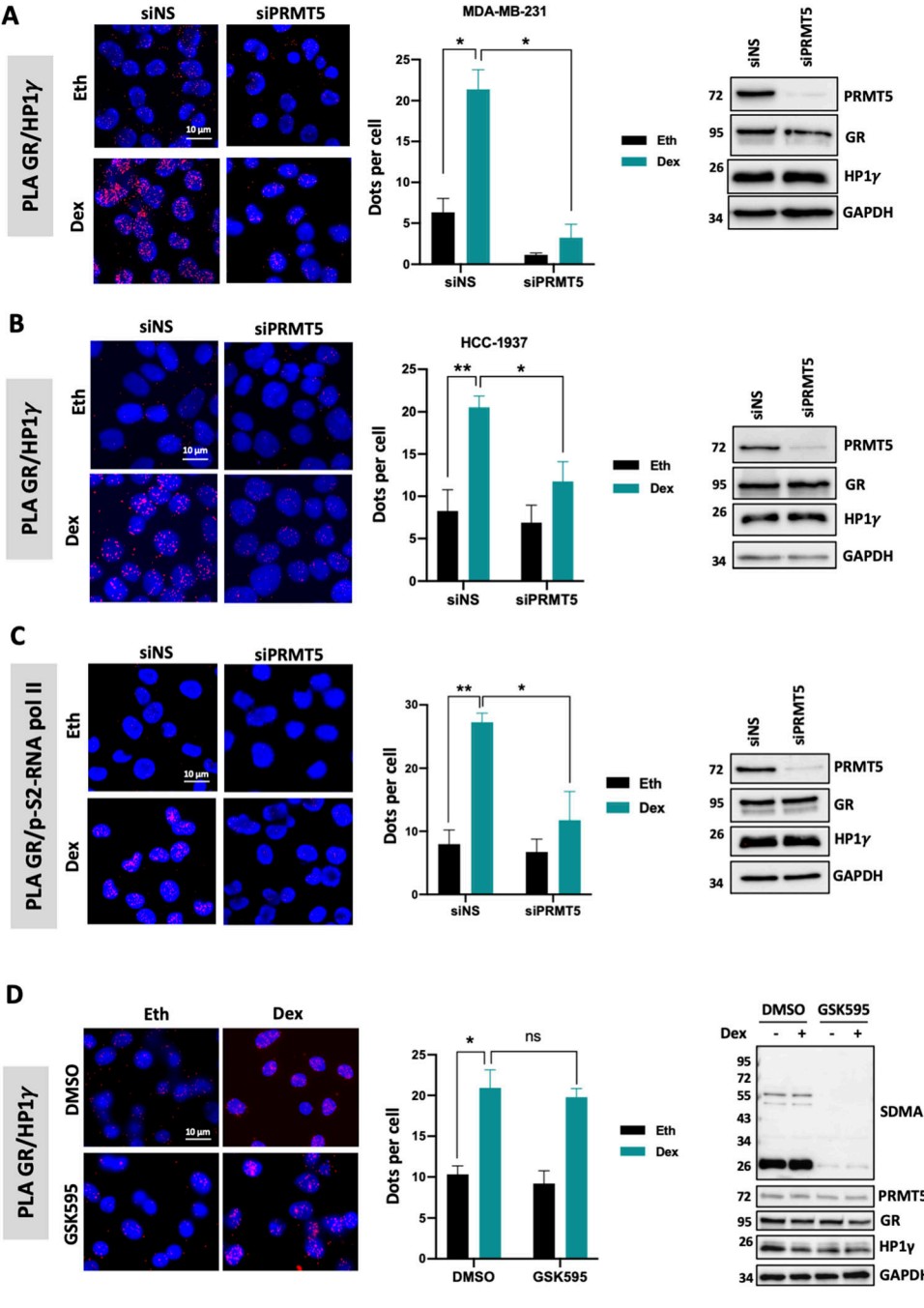

**Figure 3. PRMT5 triggers GR/HP1γ interaction.**
**(A)** PLA was conducted to analyze the interaction of endogenous GR and HP1γ after transfection of MDA-MB-231 cells with a SMART-pool siRNA targeting PRMT5 (siPRMT5) or non-specific sequence (siNS), and following treatment with 100 nM dexamethasone (Dex) or the equivalent volume of vehicle ethanol (Eth) for 2 h. The detected interactions are indicated by red dots. The nuclei were counterstained with DAPI (blue). The number of interactions in the nucleus detected by ImageJ analysis is shown as the mean ± SEM of three independent experiments. *P*-value was determined using a paired *t* test. *$P \leq 0.05$. Whole-cell extracts were analyzed for PRMT5, HP1γ, GR and GAPDH expression by immunoblot. **(A, B)** PLA was conducted as in (A) after transfection of HCC1937 cells with SMART-pool siRNA targeting PRMT5 (siPRMT5) or siNS, and following treatment with 100 nM Dex or the equivalent volume of Eth for 2 h. The number of interactions by ImageJ analysis is shown as the mean ± SEM of three independent experiments. *P*-value was determined using a paired *t* test. *$P \leq 0.05$, **$P \leq 0.01$. Whole-cell extracts were analyzed for PRMT5, HP1γ, GR and GAPDH expression by immunoblot. **(A, C)** PLA was conducted to analyze the interaction of endogenous GR and p-S2/S5-RNApol II as in (A) after transfection of MDA-MB-231 cells with SMART-pool siRNA targeting PRMT5 (siPRMT5) or siNS, and following treatment with 100 nM Dex or the equivalent volume of Eth for 2 h. The number of interactions by ImageJ analysis is shown as the mean ± SEM of three independent experiments. *P*-value was determined using a paired *t* test. *$P \leq 0.05$, **$P \leq 0.01$. Whole-cell extracts were analyzed for PRMT5, GR, HP1γ, and GAPDH expression by immunoblot. **(A, D)** PLA was conducted as in (A) after treatment of MDA-MB-231 cells with 500 nM of the PRMT5 inhibitor, GSK595, or the equivalent volume of vehicle DMSO for 6 d, and 100 nM Dex or the equivalent volume of Eth for 2 h. The number of interactions is shown using ImageJ analysis, as the mean ± SEM of three independent experiments. The *P*-value was determined using a paired *t* test. ns, non-significant; **$P \leq 0.01$. Whole-cell extracts were analyzed for PRMT5, GR, HP1γ, SDMA, and GAPDH expression by immunoblot.

subgroups, and that PRMT5 and HP1γ expression decreased in the TNBC subgroup (Fig S8B). Given our findings, we then focused on the TNBC subgroup and observed that GR mRNA expression was higher in metaplastic and apocrine TNBCs compared with unspecialized TNBCs (Fig 7A). In addition, PRMT5 and HP1γ expression slightly increased in the apocrine and metaplastic subgroups, respectively (Fig 7A). Metaplastic BC represents a rare and aggressive subtype of TNBC with a higher rate of metastasis development compared with

other TNBCs (Reddy et al, 2020). Likewise, the mRNA expression of GR was higher in PDXs engrafted from metastases in comparison to those engrafted from primary tumors (Fig 7B) and remained stable for PRMT5 and HP1γ (Fig S8C). We analyzed GR/PRMT5 interactions in TNBC PDX models including models engrafted from treatment-naïve patients (primary tumor) and TNBC tumors established from residual tumors after neoadjuvant chemotherapy administered in association with GCs (residual tumors). We observed

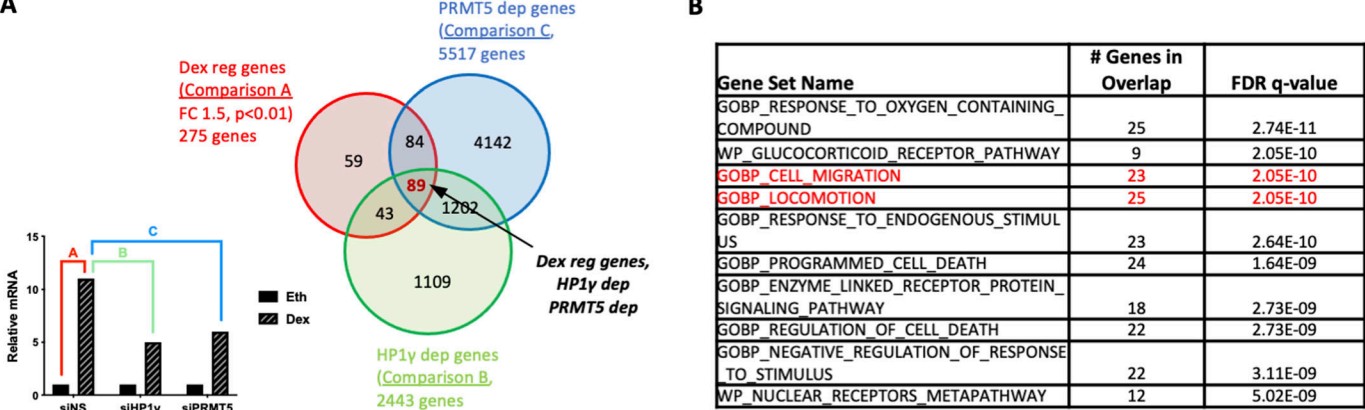

**A**

Dex reg genes (Comparison A FC 1.5, p<0.01) 275 genes

PRMT5 dep genes (Comparison C, 5517 genes)

HP1γ dep genes (Comparison B, 2443 genes)

59 — 84 — 4142 — 89 — 43 — 1202 — 1109

Dex reg genes, HP1γ dep PRMT5 dep

**B**

| Gene Set Name | # Genes in Overlap | FDR q-value |
|---|---|---|
| GOBP_RESPONSE_TO_OXYGEN_CONTAINING_COMPOUND | 25 | 2.74E-11 |
| WP_GLUCOCORTICOID_RECEPTOR_PATHWAY | 9 | 2.05E-10 |
| GOBP_CELL_MIGRATION | 23 | 2.05E-10 |
| GOBP_LOCOMOTION | 25 | 2.05E-10 |
| GOBP_RESPONSE_TO_ENDOGENOUS_STIMULUS | 23 | 2.64E-10 |
| GOBP_PROGRAMMED_CELL_DEATH | 24 | 1.64E-09 |
| GOBP_ENZYME_LINKED_RECEPTOR_PROTEIN_SIGNALING_PATHWAY | 18 | 2.73E-09 |
| GOBP_REGULATION_OF_CELL_DEATH | 22 | 2.73E-09 |
| GOBP_NEGATIVE_REGULATION_OF_RESPONSE_TO_STIMULUS | 22 | 3.11E-09 |
| WP_NUCLEAR_RECEPTORS_METAPATHWAY | 12 | 5.02E-09 |

**Figure 4. HP1γ and PRMT5 regulate a subset of GR target genes genome-wide RNA-sequencing analysis was performed on MDA-MB-231 cells to identify the genes dependent on PRMT5 and HP1γ for dexamethasone (Dex)-regulated expression.**
(A) Hypothetical results of gene expression profiles for a given gene, illustrating how specific pairwise comparisons between datasets for individual samples were performed. Each bar represents hypothetical mRNA levels from RNA-seq data for cells expressing the indicated siRNAs (PRMT5 or HP1γ or non-specific sequence) and treated for 8 h with ethanol (Eth) or Dex (100 nM). Colored letters represent pairwise comparisons performed to determine sets of genes for which mRNA levels were significantly different between the samples. For instance, comparison A = set of Dex-regulated genes (fold change ≥ 2, adjusted $P$ < 0.01), comparison B = set of HP1γ-dependent genes (adjusted $P$ < 0.05, no fold change cut-off was imposed), comparison C = set of HP1γ-dependent genes (adjusted $P$ < 0.05, no fold change cut-off was

that GR/PRMT5 interactions were significantly more frequent in residual TNBC (Fig 7C). A similar observation was made in a cohort of BC PDX including luminal B and HER2 tumors (Fig S8D). Knowing that GCs are given to patients intravenously during chemotherapy, our data obtained in a cohort of PDX models suggest that GR/PRMT5 interactions may be stimulated by GCs in vivo, impacting the management of patient side effects.

# Discussion

In the present study, we unveiled PRMT5 as a new coregulator of GR involved in cell migration in TNBC. Indeed, we demonstrated that upon Dex treatment, PRMT5 plays a key role in the transcriptional activity of GR via the recruitment of HP1γ, independently of its enzymatic activity. Our data clearly establish the GR/PRMT5/HP1γ complex as a major mediator of the effects of GCs on cell migration in vitro and in vivo (Fig 7D).

PRMT5 is the major type II methyltransferase depositing the symmetric dimethylation mark within arginine residues of proteins. Among its broad spectrum of functions, PRMT5 was shown to regulate transcription by methylating histones and transcription factors or coregulators, such as E2F1, GATA4, or RelA (Stopa et al, 2015; Chen et al, 2017; Motolani et al, 2021). Dysregulated PRMT5 expression has been described in a variety of cancers; over-expression being correlated with poor survival rates (Lattouf et al, 2019b). However, the role of PRMT5 in tumorigenesis seems to be dependent on its subcellular localization. PRMT5 was reported to be localized in the nucleus, cytoplasm and near the cell membrane (Koh et al, 2015). Our group showed that in the case of BC, nuclear PRMT5 expression in ERα+ tumors was associated with prolonged disease-free survival (Lattouf et al, 2019a; Poulard et al, 2023). In the present study, we demonstrated that in TNBC, PRMT5 interacts with GR in the nucleus upon Dex treatment where they regulate the expression of GR-target genes involved in cell migration and motility.

As PRMT5 possesses oncogenic properties in various solid cancers (Shailesh et al, 2018), industrial companies have developed specific inhibitors, with promising anti-tumoral effects (Chan-Penebre et al, 2015; Lin et al, 2019). Among PRMT5 inhibitors, GSK3326595 and JNJ64619178 are currently being assessed in the clinic. GSK3326595 phase-II clinical trials for BC and acute myeloid leukemia are ongoing (Wu et al, 2021). As we showed that PRMT5 is required for the interaction between GR and HP1γ, we hypothesized that its catalytic activity could be involved, particularly because PRMT5 was described to methylate members of the nuclear receptor family (Malbeteau et al, 2022). Our team recently demonstrated that

PRMT5 triggers GR methylation, although the functional consequences have not yet been unveiled (Poulard et al, 2020). However, in the present study, we demonstrated that the PRMT5 inhibitor GSK3326595 does not affect (i) GR and HP1γ interaction, (ii) Dex-regulated target genes (PRMT5 and HP1γ dependent), and (iii) cell migration induced by GCs. These observations clearly demonstrate that the role of PRMT5 in GC-induced cell migration is not due to its enzymatic activity but that it may act as a scaffold coregulator of GR, participating in its transcriptional activation, emphasizing a new role for PRMT5 independently of its well-known methyltransferase activity. Another way for targeting PRMT5 could be via the proteolysis-targeting chimera technology. This technology, triggering PRMT5 degradation, could be an opportunity to target the oncogenic activity of GR in TNBC through the PRMT5-scaffolding capacity (Shen et al, 2020). The proteolysis-targeting chimera PRMT5 inhibitor (MS4322) is a valuable chemical tool for targeting both the catalytic activity of PRMT5 and its scaffolding capacity. Preliminary data on MS4322 demonstrated a good plasma exposure in mice, indicating that MS4322 could potentially be transposed to clinical trials (Shen et al, 2020). It will be an asset for targeting cell migration induced by GCs highlighted in our current work and could be used for other potential applications. Indeed, similar observations were made for the role of PRMT5 in vascular morphogenesis (Quillien et al, 2021). Authors found that the catalytic activity of PRMT5 was required for blood cell formation but not for vessel formation by promoting proper chromatin conformation.

Most coregulators were discovered for their role in either transcriptional activation or repression; by definition, coregulators that help activate genes are called coactivators, and corepressors involved in the repression of transcription. However, many coregulators function in both activation and repression of transcription, depending on the specific gene and the cellular environment. Several reports demonstrated that this switch could involve post-translational modifications. Likewise, the lysine methyltransferases G9a and GLP not only catalyze the methylation of H3K9, a well-known repressive mark, but can also act as a coactivators of GR, ERα, and other transcription factors (Chaturvedi et al, 2009; Purcell et al, 2011; Bittencourt et al, 2012). Recent data showed that the coactivating activity of G9a/GLP is modulated by a methylation/phosphorylation switch (Poulard et al, 2017). The coactivating function requires G9a/GLP self-methylation to provide a binding site for the coregulator HP1γ, which is required as a cooperating coactivator for G9a and GLP. In contrast, G9a/GLP phosphorylation of the threonine adjacent to the methylation site by Aurora kinase B prevents binding to HP1γ and reduces the coactivating function of G9a and GLP. Unlike findings in the lung adenocarcinoma cell line A549 and B-cell acute lymphoblastic leukemia Nalm6, G9a and GLP are not involved in GR/HP1γ interactions in TNBC cells, highlighting that GR transcription factors regulate

imposed). Venn diagram was obtained using these comparisons. Overlap area (89 genes in red) indicates the number of genes shared among sets. **(B)** Gene Ontology Analysis using GSEA identifies Dex-regulated gene networks dependent upon PRMT5 and HP1γ. Gene sets are ranked according to their normalized enrichment score. The false discovery rate is the estimated probability that a gene set with a given normalized enrichment score represents a false-positive. **(C)** MDA-MB-231 cells transfected with non-specific sequence or with SMART-pool siRNA targeting PRMT5 (siPRMT5) or HP1γ (si HP1γ) were treated with 100 nM Dex or the equivalent volume of Eth for 8 h. mRNA levels for the indicated GR target genes were measured by reverse transcriptase followed by qPCR and normalized against 28S mRNA levels. Results shown are mean ± SEM of four independent experiments. $P$-value was calculated using a paired $t$ test. *$P \leq 0.05$, **$P \leq 0.01$, ***$P \leq 0.001$. **(C, D)** mRNA levels for the indicated GR target genes were determined as in (C), MDA-MB-231 cells were treated with 500 nM of the PRMT5 inhibitor, GSK595, or the equivalent volume of vehicle DMSO for 48 h, and then with 100 nM Dex or the equivalent volume of Eth for 8 h. Results shown are mean ± SEM of four independent experiments. $P$-value was calculated using a paired $t$ test. ns, non-significant, **$P \leq 0.01$.

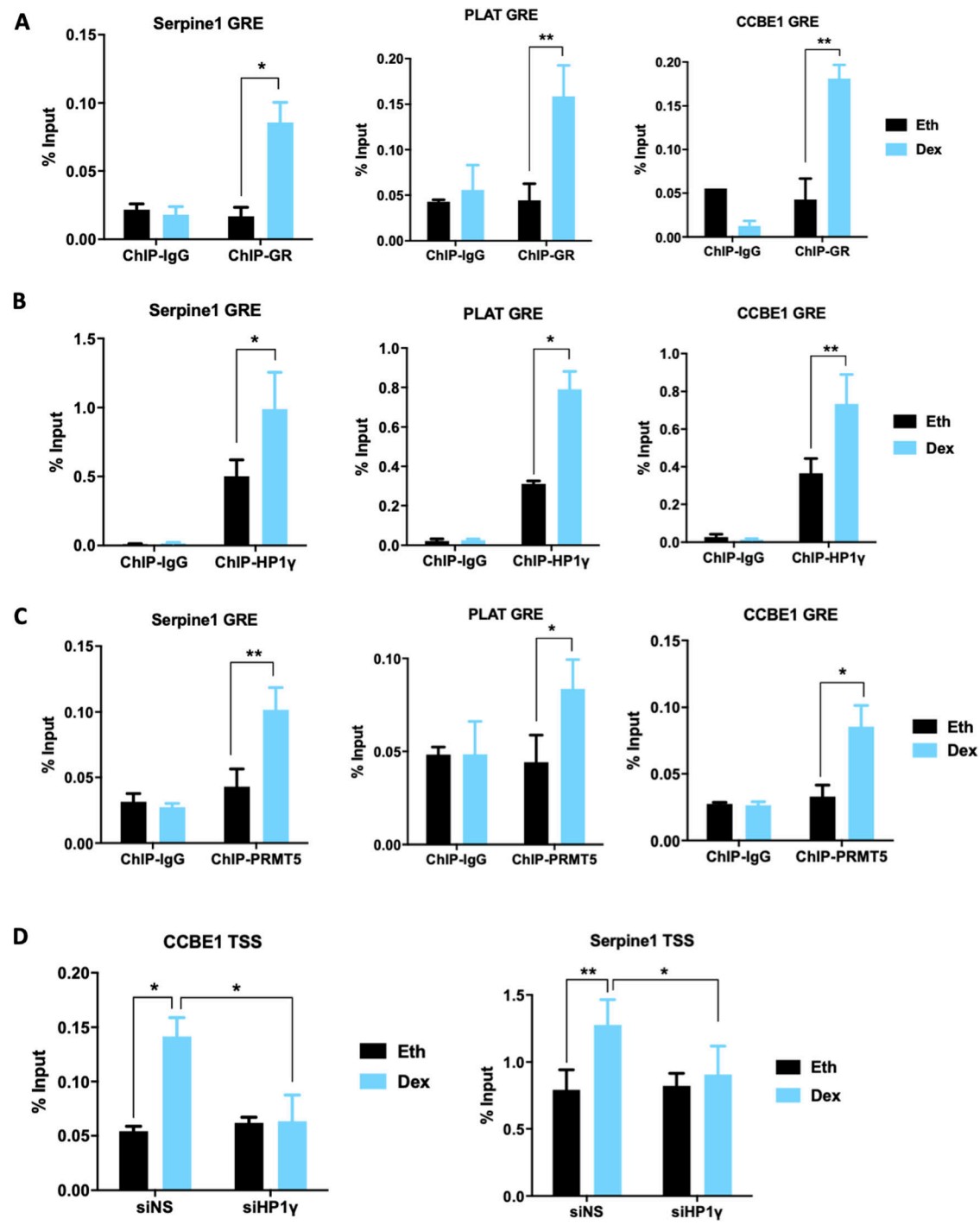

**Figure 5. Occupancy of GR, HP1γ, and PRMT5 on GR response elements of GR target genes.**
**(A, B, C)** MDA-MB-231 cells were treated with 100 nM dexamethasone (Dex) or an equivalent volume of vehicle ethanol (Eth) for 2 h. **(A, B, C)** ChIP was performed with (A) GR, (B) HP1γ, or (C) PRMT5 antibodies and immunoprecipitated DNA was analyzed by qPCR using primers specific for the GR-binding region associated with the indicated genes. Results are normalized against input chromatin, and the mean ± SEM of the ratio between 2 h Dex or Eth treatment for three independent experiments is shown. P-value was calculated using a paired t test. *P ≤ 0.05, **P ≤ 0.01. **(D)** MDA-MB-231 cells were transfected with non-specific sequence or with SMART-pool siRNA targeting HP1γ (siHP1γ) and treated with 100 nM Dex or Eth for 2 h. ChIP was performed with an antibody against RNA polymerase II phosphorylated on S2 and S5 of the C-terminal domain repeats (p-S2/S5-RNApol II), and immunoprecipitated DNA was analyzed by qPCR using primers that amplify the transcription start site associated with the indicated GR target genes. Results are normalized against input chromatin and shown as mean ± SEM of three independent experiments. P-value was calculated using a paired t test. *P ≤ 0.05, **P ≤ 0.01.

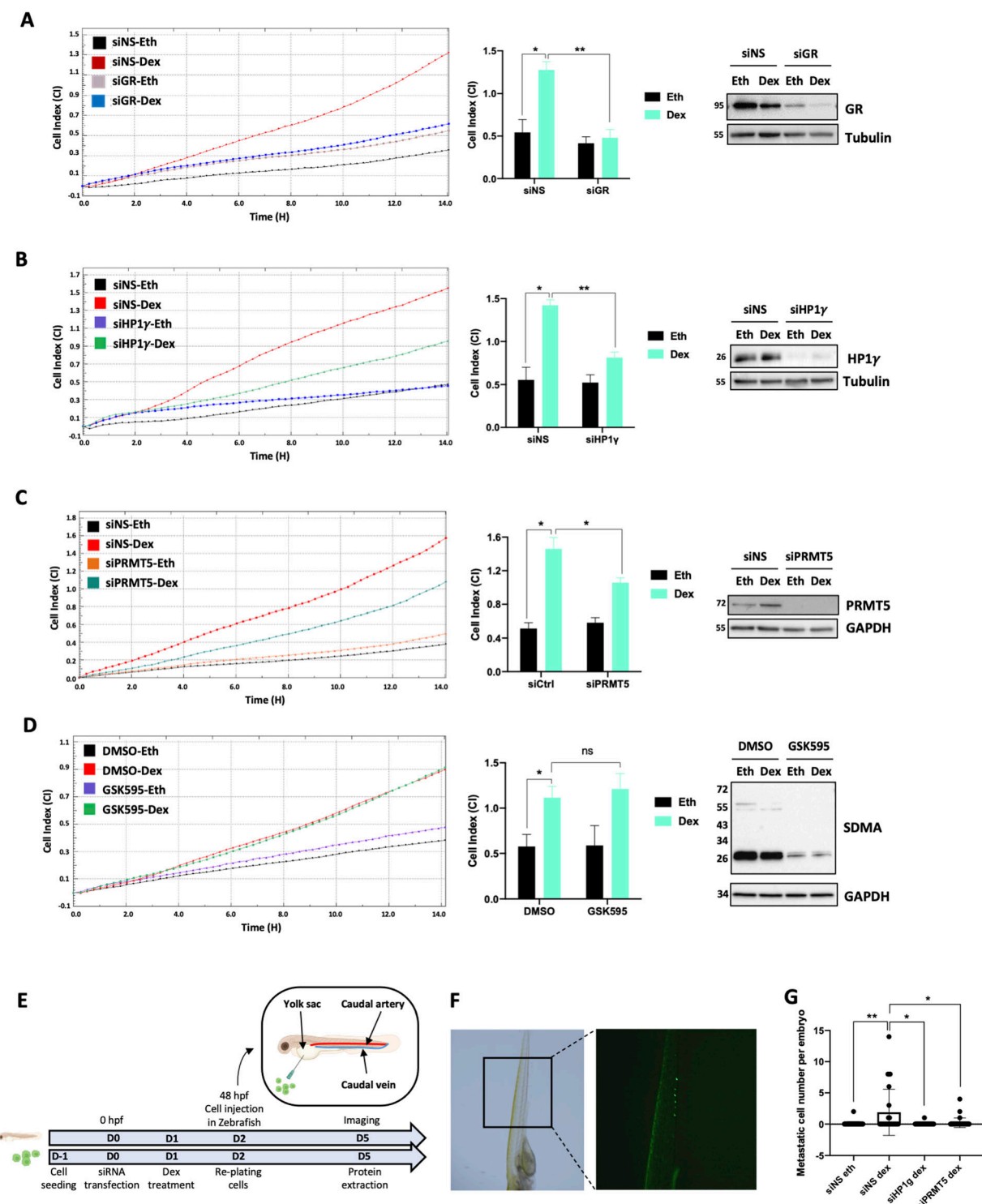

**Figure 6. HP1γ and PRMT5 regulate cell migration induced by dexamethasone (Dex).**
**(A)** MDA-MB-231 cells were transfected with non-specific sequence (siNS) or with SMART-pool siRNA targeting GR (siGR) and treated with 100 nM Dex or an equivalent volume of ethanol (Eth) for 24 h. 40,000 cells were seeded per well in the 16-well Real-Time Cell Analyzer plate. Cell index (CI) values are presented as means ± SD of at least two (up to three) independent wells, calculated by xCELLigence. Graph of one representative experiment is shown (left panel). Histogram showing the results of three independent experiments, and P-value was calculated using a paired t test. *P ≤ 0.05, **P ≤ 0.01. Whole-cell extracts were analyzed for GR and tubulin expression by immunoblot. **(A, B)** xCELLigence was performed and analyzed as in (A) after transfection of MDA-MB-231 with siNS or with SMART-pool siRNA targeting HP1γ (siHP1γ) and treated with 100 nM Dex or Eth for 24 h. Whole-cell extracts were analyzed for HP1γ and tubulin expression by immunoblot. **(A, C)** xCELLigence was performed and analyzed as in (A) after transfection of MDA-MB-231 with siNS or with SMART-pool siRNA targeting PRMT5 (siPRMT5) and treated with 100 nM Dex or Eth for 24 h. Whole-cell extracts were analyzed for PRMT5 and GAPDH expression by immunoblot. **(A, D)** xCELLigence was performed and analyzed as in (A) after treatment of MDA-MB-231 with 500 nM of

functions in a tissue-dependent manner through the recruitment of different sets of coregulators (Poulard et al, 2017, 2019). In TNBC, we showed that HP1γ is recruited to GR through PRMT5 (Fig 7D). In addition, we demonstrated that HP1γ acts as a coactivator of GR through its phosphorylation. Identifying the kinase that catalyzes HP1γ phosphorylation on S93 residue in TNBC could be another way to target the GR/PRMT5/HP1γ complex driving GC-induced migratory properties.

As the formation of the GR/HP1γ/PRMT5 complex is induced by GC treatment, we investigated the molecular mechanisms associated with the deleterious effects of GR in this context. Our RNA-seq analysis demonstrated that PRMT5 is a master coregulator of GR as it regulates 63% of Dex-regulated genes in a large-scale analysis. In comparison, HP1γ regulates 48% of Dex-regulated genes. Among the Dex-regulated genes, 32% required both HP1γ and PRMT5. Interestingly, we showed by GO analysis that these genes are enriched in cell migration and locomotion pathways. For instance, SERPINE1 is a protein that promotes cytoskeletal rearrangement driving cellular migration, actin-rich migratory structures, and reduced actin stress fibers (Humphries et al, 2019). We also demonstrated that this GR/PRMT5/HP1γ complex drives the migratory properties induced by GCs in TNBCs both in vitro in human TNBC cells and in vivo in the zebrafish model.

Finally, our study demonstrates that GR/PRMT5 interaction should not be assessed at diagnosis as this interaction remains weak and constant among tumors. However, our study demonstrated that upon chemotherapy supplemented with GCs, GR/PRMT5 interaction increased in some tumors leading to potential risk of metastatic progression. Indeed, GR/PRMT5 interaction is higher in TNBC tumors treated with chemotherapy and GCs before engraftment and showing a partial (or no) response to chemotherapy. The residual tumors of these patients were taken at surgery and engrafted in mice. These data suggest that, in vivo, GR/PRMT5 interactions could be stimulated by GCs, though the cause of this enhanced interaction remains unknown as it could be due to the combination of GCs and chemotherapy or to the resistance mechanism itself, as residual tumors are resistant to chemotherapy. In addition, this study is a proof-of-concept that targeting GR/PRMT5/HP1γ complex formation could prevent the development of metastases in TNBC patients.

Early TNBC patients treated with chemo-immunotherapy combination in the neoadjuvant setting often need long-term steroid treatments to reduce the adverse effects from the multi-drug regimen. However, the prognostic impact of this GC intake on survival has been poorly explored. Given our data and the potential risk of metastatic progression, the routine premedication prescription of GCs for early TNBC patients has to be further assessed in prospective clinical trials for its reevaluation.

# Materials and Methods

## Cell culture

MDA-MB-231, BT549, BT20, HCC-1937, MDA-MB-453, and Cos-7 cells were cultured with specific medium and 10% FBS at 37°C and in 5% $CO_2$. Before experiments, cells were grown in phenol red-free medium supplemented with 10% charcoal-stripped serum (Biowest).

When indicated, cells were treated with 100 nM Dex (Sigma-Aldrich), 100 nM prednisolone (Selleckchem), 1 μM hydrocortisone (Selleckchem), 1 μM RU486 (Selleckchem), or with 0.5 μM PRMT5 inhibitor GSK3326595 called GSK595 (Selleckchem) for the indicated time.

SMART-pool siRNAs (Dharmacon) used for the depletion of GR, HP1γ, PRMT5, G9a, GLP, and siNS were transfected into indicated cells using Lipofectamine siRNAi max (Invitrogen) according to the manufacturer's protocol.

## PLAs

The experiments were performed using reagents from the PLA Kit (DUO92004, DUO92002, DUO92007, DUO82049, DUO82040; Sigma-Aldrich) as previously described (Poulard et al, 2020). Cells were seeded onto coverslips in 12-well plates, fixed in methanol for 2 min, and then washed twice in 1X-PBS. Fixed cells were stored at 4°C for subsequent staining or saturated with the blocking solution for 1 h at 37°C. Cells were then incubated with different pairs of primary antibodies (GR [sc-393232; Santa Cruz], HP1γ [ab10480; Abcam], p-S93-HP1γ [ab45270; Abcam], PRMT5 [07-405; Millipore], and p-S2/S5-RNApolII [#4735; Cell signaling]) for 1 h at 37°C. After three washes in Buffer A, the PLA minus and plus probes which contain the secondary antibodies conjugated with complementary oligonucleotides were added and incubated for 1 h at 37°C. Again, cells were washed three times in Buffer A and incubated with T4 DNA ligase in diluted ligase buffer for 30 min at 37°C. Subsequently, after three washes with Buffer A, cells were incubated with DNA polymerase in dilution polymerase buffer containing red fluorescent-labeled oligonucleotides for 100 min at 37°C. Finally, cells were washed twice with 1X-Buffer B for 10 min at RT, then 1 min with 0.01X Buffer B. The samples were mounted using Duolink in situ mounting medium containing DAPI. The edges of the coverslips were sealed using nail polish. Slides were then be stored in the dark at 4°C for a short period of time or visualized under a Nikon Fluorescence Microscope, and interactions were counted using ImageJ software. For each sample, interactions were counted for at least 1,000 cells using ImageJ software (Poulard et al, 2020). As described in Poulard et al (2020), combining the values of the number of dots and the number of cells, we obtained an estimation of the number of dots per cell in a given condition. As

---

GSK595, or the equivalent volume of vehicle DMSO for 72 h and treated with 100 nM Dex or ethanol for 24 h. Whole-cell extracts were analyzed for sDMA and GAPDH expression by immunoblot. **(E)** Working diagram of the zebrafish model. MDA-MB-231 were transfected with siNS, with SMART-pool siRNA targeting PRMT5 (siPRMT5) or with SMART-pool siRNA targeting HP1γ (siHP1γ), and treated with Dex 24 h before transplantation. MDA-MB-231 cells were labeled using DiO at the injection time and injected into the yolk sac of 2-d old zebrafish embryos. Larvae were imaged with a fluorescent microscope 3 d post-transplantation. Hpf, hours post-fertilization. **(F)** A representative epifluorescence image of the caudal blood vessels shows invasion of cancer cells. **(G)** Quantification of invaded metastatic cells per embryo under different conditions. *P*-value was calculated using unpaired *t* test *$P \leq 0.05$, **$P \leq 0.01$.

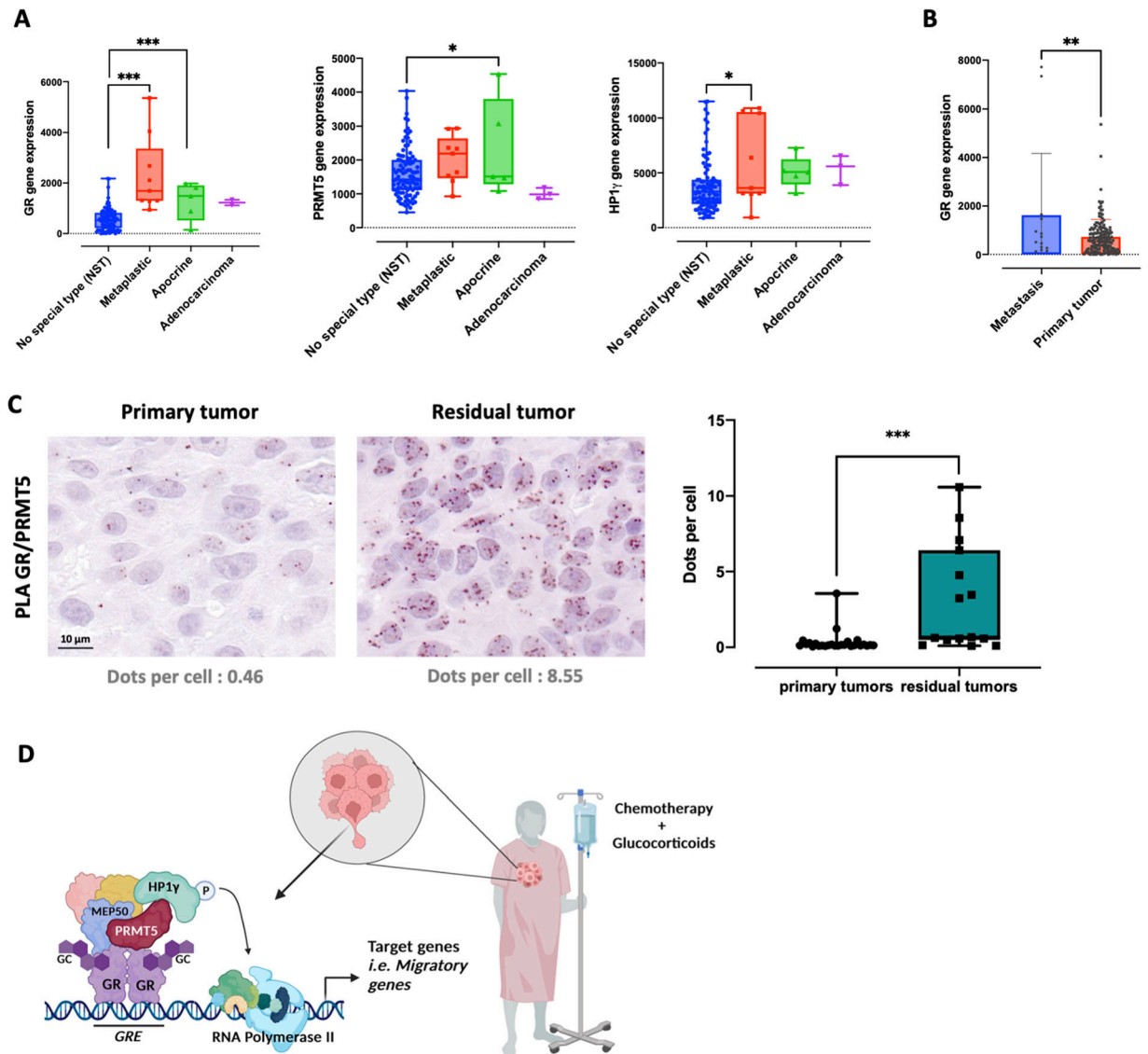

**Figure 7. Study of GR and GR/PRMT5 expression in breast cancer patients.**
**(A)** GR, PRMT5, and HP1γ expression was analyzed in the triple-negative breast cancer (TNBC) subtype divided into different clinical subtypes. The *P*-value was calculated using unpaired *t* test. *$P ≤ 0.05$, ***$P ≤ 0.001$. **(B)** GR expression was analyzed in patient-derived xenografts engrafted from metastatic tumors or primary tumors. The *P*-value was calculated using unpaired *t* test **$P ≤ 0.01$. **(C)** GR/PRMT5 interaction was analyzed in 40 TNBC patient-derived xenografts by PLA. Primary tumors: no treatment before engraftment. Residual tumors: tumors treated by chemotherapy and GCs with a partial or no response before engraftment. Two examples of different staining profiles are shown (Obj: X40). **(D)** Model of PRMT5 role in TNBC following GC treatment. Created with BioRender.com (Agreement number BV24ZV750Y).

images were acquired for at least eight randomly chosen fields of view in an automated manner and combined with an image analysis macro designed to apply the same criteria to all of the pictures, this reduced the effect of human input/error on the values. In addition, analysis of at least 300 cells per condition increased data reliability. The analysis methods was previously described (Poulard et al, 2020).

### GST pull-down experiments

psg5-V5-PRMT5 and pcdna3.1-GR expressing plasmids were transcribed and translated using in vitro T7-coupled reticulocyte lysate.

GST, GST-HP1γ and GST-PRMT5 proteins were incubated with labeled proteins in 200 µl binding buffer (Tris 20 mM pH 7.4, NaCl 0.1 M, EDTA 1 mM, glycerol 10%, Igepal 0.25% with 1 mM DTT and 1% milk) for 2 h at RT. After washing, bound proteins were separated by SDS–PAGE and visualized by Western blot.

### Immunoprecipitation and Western blot analysis

Cos-7 cells were seeded onto 10-cm² dishes the day before transfection. The following plasmids psg5-V5-PRMT5, pcdna3.1-HA-HP1γ, and pcdna3.1-GR were transfected into Cos-7 cells using Lipofectamine 2000 (Invitrogen) according to the manufacturer's protocol. 48 h

**Table 1. List of primers used in qPCR.**

| Primer | Forward sequence (5′-3′) | Reverse sequence (5′-3′) |
|---|---|---|
| CCBE1 | GCCTTGCTTAATGTGGGACA | CACCAGGACCAAAGGGAAG |
| HP1γ | ACTGCCATCACAGCAGGTTT | CTAAGGAATGGCCCGCTAGG |
| IGFBP3 | AACTTTGTAGCGCTGGCTGT | TGCTAGTGAGTCGGAGGAAGA |
| PLAT | GAGAATCCAGCAGGAGCTGA | AGACAGTACAGCCAGCCTCA |
| PRMT5 | CGGGGACTGCAGATAGTCTT | GTGCAGTTCATCATCACAGG |
| SERPINE1 | TCTTTGGTGAAGGGTCTGCT | CTGGGTTTCTCCTCCTGTTG |
| 28S | CGATCCATCATCCGCAATG | AGCCAAGCTCAGCGCAAC |

after transfection, cells were treated (or not) with Dex for 24 h, and cell extracts were prepared in RIPA buffer (50 mM Tris–HCl, pH 8.0, 150 mM NaCl, 1 mM EDTA, 1% NP-40, and 0.25% deoxycholate) supplemented with protease inhibitor tablets and phosphatase inhibitors (1 mM NaF, 1 mM $Na_3VO_4$, and 1 mM $β$-glycerophosphate). Protein extracts were incubated with HP1γ primary antibody (ab10480; Abcam), PRMT5 (ab109451; Abcam) or GR/DGH2L (#12041; Cell signaling) over night at 4°C under agitation. Protein A Agarose (Millipore) beads were then added, and the mixture was incubated 2 h at 4°C. The immunoprecipitates were separated on SDS–PAGE. Immunoblotting was conducted with primary antibodies against GR G-5 (sc-393232; Santa Cruz), GR/D6H2L (#12041; Cell signaling), HP1γ (ab10480; Abcam), and PRMT5 (07-405; Millipore). Secondary antibodies were used for chemiluminescence detection using the ECL detection reagent (Roche Molecular Biochemicals) according to the manufacturer's instructions. For immunoprecipitation experiments, 3% of the input of each sample was analyzed by immunoblot.

## RNA extraction and RT-qPCR analysis

Total RNA was extracted using TRIzol (Invitrogen) in accordance with the manufacturer's instructions. cDNA was synthesized by reverse transcription using SuperScript III (Invitrogen) according to the manufacturer's specifications with 1 $μg$ total RNA as a template. Quantitative PCR amplification of the resulting cDNA was performed on a Bio-Rad CFX real-time PCR system using SYBR Green Supermix (Bio-Rad). mRNA levels were normalized against the level of 28S mRNA. For amplification of cDNA, primer sequences are listed in Table 1.

## Chromatin immunoprecipitation

ChIP assays were performed using the "Simple ChIP Plus Enzymatic Chromatin IP Kit" (Cell Signaling) according to the protocol described. Briefly, MDA-MB-231 cells were transfected with siNS or siHP1γ when specified and were subjected to hormonal treatment (100 nM Dex) for 2 h. Cells were then cross-linked using 1% formaldehyde (Sigma-Aldrich) in 15 cm in diameter culture dishes containing 20 ml medium and incubated for 10 min at RT. Then, 2 ml of 10X glycine (Cell signaling) was added to each 15 cm in diameter dish and cells were further incubated for 5 min at RT to stop the cross-linking. Cell extracts were then prepared, and chromatin digested and sonicated. Immunoprecipitation of sonicated chromatin solutions was conducted overnight at 4°C with the following antibodies: normal rabbit IgG (#2729; Cell Signaling Technology), anti-GR/D6H2L (#12041; Cell Signaling Technology), anti-

HP1γ (ab10480; Abcam), anti-PRMT5 (07-405; Sigma-Aldrich), and p-S2/S5 RNA pol II (#4735; Cell signaling Technology). Cross-linking was reversed by heating, and immunoprecipitated DNA was purified and analyzed by qPCR as described above. Results are expressed relative to the signal obtained from input chromatin. Primer sequences are indicated in Table 2.

## RNA sequencing

RNA-sequencing experiments were performed using MDA-MB-231 cells. Cells were transfected with siNS, siPRMT5, and siHP1γ (25 nM) for 48 h and treated with Dex (100 nM) for 8 h before RNA extraction. A total of 18 high-quality samples (six conditions × three replicates each) were submitted to the IGFL (Institute of Functional Genomic of Lyon) sequencing platform for library preparation and sequencing. cDNA libraries were prepared using the RNA-seq library prep kits with UDIs (Lexogen). All libraries were sequenced on an Illumina Nextseq500 and mapped on the hg38 version of the human genome using Bowtie2 (Galaxy Version 2.4.2 Galaxy 0). Count tables were prepared using htseq-count (Galaxy version 0.9.1). Differential gene expression analysis was performed with DESeq2 (Galaxy Version 2.11.40.7 galaxy1) using different thresholds. RNA-sequencing data have been submitted to the Gene Expression Omnibus GSE237596.

## xCELLigence analysis

The Roche xCELLigence Real-Time Cell Analyzer DP instrument was used to monitor and record real-time cellular migration without labeling cells. xCELLigence assays were performed using a CIM (cellular invasion/migration)-Plate 16 (Agilent) which contains microelectronic sensors integrated to the underside of the microporous polyethylene terephthalate membrane of a Boyden-like chamber, in accordance with the manufacturer's guidelines. First, 160 $μl$ of complete red medium containing 100 nM Dex or its vehicle ethanol were added to the lower chamber of the CIM-plate and placed for 1 h in a $CO_2$ incubator at 37°C. Then, MDA-MB-231 cells transfected with siNS, siGR, siHP1γ, or siPRMT5 or treated with 500 nM of GSK595 for 72 h and subjected to hormonal treatment (100 nM Dex) for 24 h before assay were trypsinized, resuspended, and counted. Next, 150 $μl$ of complete red medium containing ~40,000 MDA-MB-231 cells and 100 nM Dex was added to the upper chamber of the CIM-Plate. The CIM-Plates were assembled by placing the top chamber onto the bottom chamber and placed for 30 min in the $CO_2$ incubator at 37°C to let cells settle down. The CIM-Plate was placed into the xCELLigence analyzer and incubated for 24 h at 37°C and 5% $CO_2$. Cells migrating from the upper chamber through the polyethylene terephthalate membrane to the lower chamber in response to Dex adhere to electronic sensors, resulting in an increase in impedance. Increased impedance is correlated with an increased number of cells migrating, and cell index (CI) values reflecting the changes in impedance were automatically recorded every 15 min, and the time point closer to 14 h (above or below) was used for data analysis.

## Human breast cancer sample collection

The tumors of 442 patients of the Centre Léon Bérard with invasive BC sampled at diagnosis, whose clinical and biological data were available from the regularly updated institutional database, were analyzed.

**Table 2. List of primers used for ChIP experiments.**

| Primer | Forward sequence (5′-3′) | Reverse sequence (5′-3′) |
|---|---|---|
| CCBE1 GRE | CCCTGGTTGAAGGAAAGGAT | ATGTTGGGTACCAACCCTCA |
| CCBE1 GRE (2) | TCCACTGATAGGGGCAAAA | CAGGAAGGTCCGTGGTAAT |
| CCBE1 TSS | GGGGAAAATGAGGCTAGGA | TCCAGCAAGTCTGTCAATCG |
| PLAT GRE | CTTTGGGAGAGCGGCCAAAG | CGAGTCCTGTGATGCCATGG |
| SERPINE1 GRE | GAGAGATCGCTGTGGTCCAT | GTGCAAAGGAGGAGAGATC |
| SERPINE1 GRE-TSS | CAGAGGGCAGAAAGGTCAA | CTCTGGGAGTCCGTCTGAA |

**Table 3. List of primers used in qPCR for patient-derived xenografts samples.**

| Primer | Forward sequence (5′-3′) | Reverse sequence (5′-3′) |
|---|---|---|
| TBP | AGAACAACAGCCTGCCACCTTAC | GGGAGTCATGGCACCCTGAG |
| GR (NR3C1) | GCCAAGGATCTGGAGATGACAACT | GGTCTCATGCTGGGGCTTGAA |
| PRMT5 | AATGCCGTGGTGACGCTAGA | CATGTCTGATGAGACTACGGTCACTT |
| HP1γ (CBX3) | AGGCAGAGCCTGAAGAATTTGTC | TTCCCAAGTATTGTCAGCATCTGTA |

Written informed consent was obtained from each patient. The study protocol was approved by the Institutional Ethics Committee.

### PDX tumors

PDX models of breast cancer were established from early stage breast cancers as previously described (Marangoni et al, 2007; Coussy et al, 2019).

### RNA extraction and RT–PCR analysis of *GR*, *PRMT5*, and *HP1γ* in PDX

RNA extraction was performed by using acid–phenol guanidium method. Electrophoresis through agarose gel staining with ethidium bromide was performed to determine the RNA quality, and 18S and 28S RNA bands were visualized under ultraviolet light. RNA was reverse transcribed in a final volume of 20 $\mu$l containing 1× RT buffer (500 mm each dNTP, 3 mm $MgCl_2$, 75 mm KCl, and 50 mm Tris–HCl [pH 8.3]), 10 U of RNasinTM RNase inhibitor (Promega), 10 mm DTT, 50 units of Superscript II RNase H-reverse transcriptase (Life Technologies, Inc.), 1.5 mm random hexamers (Pharmacia), and 1 $\mu$g of total RNA. The samples were incubated at 20°C for 10 min and 42°C for 30 min, and reverse transcriptase was inactivated by heating at 99°C for 5 min and cooling at 5°C for 5 min. For amplification of cDNA, primer sequences are listed in Table 3.

We used protocols for PCR amplification of the *GR* (*NR3C1*), *PRMT5*, and *HP1γ* (*CBX3*) genes described in detail elsewhere (Tozlu et al, 2006). Briefly, we obtained quantitative values from the Ct value (cycle number) at which the increase in the fluorescence signal associated with exponential growth of PCR products was detected by the laser detector of the QuantStudio 7 sequence detection system (Perkin-Elmer Applied Biosystems), using according to the manufacturer's manuals (QuantStudio Real-Time PCR Software v1.5).

The human TATA box-binding protein (*TBP*, GenBank accession no. NM_003194) gene was used for gene normalization. Results,

expressed as N-fold differences in *GR* expression relative to the *TBP* gene and termed "N*GR*," were calculated as N*GR* = $2^{\Delta Ctsample}$, where the ΔCt value was determined by subtracting the average Ct value of *GR* gene from the average *TBP* gene Ct value.

### In vivo dissemination assay in zebrafish larvae

Immediately before transplantation, MDA-MB-231 cells were labeled with DiO fluorescent dye (Invitrogen Molecular Probes) according to manufacturer's instructions and resuspended in PBS at a final concentration of 60,000 cells/$\mu$l. 2-d-old zebrafish larvae of the *casper* strain were anesthetized with tricaine (MS-222). 10 nl containing ~300 MDA-MB-231 cells were injected into the middle of the yolk sac with a microinjector. After transplantation, larvae were recovered in E3 medium and incubated at 34°C. Viable larvae with fluorescent signal in the yolk sac were sorted 6–10 h post-transplantation and transferred to individual wells of a 24-well plate containing E3 medium. Plates were incubated at 34°C. 3 d post-transplantation, larvae were imaged with a Nikon SMZ18 fluorescent stereoscope.

## Supplementary Information

## Acknowledgements

We deeply thank C Languilaire, F Nasri, and S Ensenlaz for technical support. We also thank B Manship and MR Stallcup for proofreading the article and constructive discussions. This work was supported by the Fondation ARC Cancer, "La Ligue contre le Cancer," the association "Cancer du sein parlons en," and ITMO Cancer of Aviesan within the framework of the 2021–2030 Cancer Control Strategy, on funds administered by Inserm. LM Noureddine

was supported by a fellowship from AZM and Saadeh Association and the Lebanese University. We acknowledge the Institut Convergence Plascan (Grant Number ANR-17-CONV-0002) for their support. TH Pham was supported by Campus France. This work was also supported by INCA and DGOS for the PRTK ISOTHER.

## Author Contributions

LM Noureddine: formal analysis, validation, investigation, visualization, and methodology.
J Ablain and A Surmieliova-Garnès: formal analysis, investigation, and methodology.
J Jacquemetton: software, formal analysis, and investigation.
TH Pham: investigation.
E Marangoni: resources, investigation, and methodology.
A Schnitzler and I Bieche: formal analysis, validation, and investigation.
B Badran: supervision and funding acquisition.
O Trédan: conceptualization, resources, funding acquisition, and writing—review and editing.
N Hussein: supervision and funding acquisition.
M Le Romancer: conceptualization, supervision, funding acquisition, methodology, and writing—review and editing.
C Poulard: conceptualization, formal analysis, supervision, funding acquisition, validation, methodology, and writing—original draft.

## Conflict of Interest Statement

The authors declare that they have no conflict of interest.

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
