## [Reviewer comments · Life Science Alliance]

Life Science Alliance

PRMT5 triggers glucocorticoid-induced cell migration in Triple Negative Breast Cancer

Lara Noureddine, Julien Ablain, Austra Surmieliová-Garnès, Julien Jacquemetton, Thuy Ha Pham, Elisabetta Marangoni, Anne Schnitzler, Ivan Bièche, Bassam Badran, Olivier Tredan, Nader Hussein, Muriel Le Romancer, and Coralie Poulard

DOI: <https://doi.org/10.26508/lsa.202302009>

Corresponding author(s): Coralie Poulard, Cancer Center of Lyon and Muriel Le Romancer, CRCL

Review Timeline:

Submission Date:	2023-02-23
Editorial Decision:	2023-03-30
Revision Received:	2023-06-30
Editorial Decision:	2023-07-14
Revision Received:	2023-07-18
Accepted:	2023-07-24

Transaction Report:

March 30, 2023

Re: Life Science Alliance manuscript #LSA-2023-02009-T

Dr. Coralie Poulard
CRCL
Inserm U1052 CNRS UMR 5286
Centre Léon Bérard, 28 rue Laennec
Lyon 69373 Lyon Cedex 08
France

Dear Dr. Poulard,

Thank you for submitting your manuscript entitled "PRMT5 triggers glucocorticoid-induced cell migration in Triple Negative Breast Cancer" to Life Science Alliance. The manuscript was assessed by expert reviewers, whose comments are appended to this letter. We invite you to submit a revised manuscript addressing the Reviewer comments.

Thank you for this interesting contribution to Life Science Alliance. We are looking forward to receiving your revised manuscript.

Sincerely,

B. MANUSCRIPT ORGANIZATION AND FORMATTING:

Reviewer #1 (Comments to the Authors (Required)):

Triple negative breast cancer is currently treated using chemotherapy and immune modulators. Synthetic glucocorticoids (GCs) are given to patients to manage the side effects of these treatments. However, the use of GCs is now being questioned because of correlation with metastatic disease. In this paper, Nouredine et al., investigate the binding partners of glucocorticoid receptor (GR) in TNBC cell lines to understand how GCs mechanistically promote cell migration. They find that GR forms a complex with PRMT5 and HP1, and that either component is required for GR-mediated chromatin localisation and gene expression, and cell migrations. Finally, using patient samples and tumours derived from breast PDX models, they very nicely demonstrate that the PRMT5-GR interaction is enhanced in chemotherapy treated rather than naive tumours. As such, the PRMT5-GR-HP1 complex could be prognosis for metastatic disease.

This study addresses an important clinical question and could have significant translational impact. The experiments are well designed and the data of a very high standard and I very much enjoyed reading the manuscript. A minor criticism is the heavy reliance of PLA for protein interactions in cell lines, however the adaptation of this for FFPE tissue is an exciting development. I have a few comments:

1. I appreciate the controls put in place to determine that the PLA signal is not artefactual, however it would be good to see key cell line data supported by co-immunoprecipitation studies. The authors IP HP1 and detect PRMT5 and GR interactions (Fig 2D), can this be repeated with reciprocal co-IPs? If this is not possible due to antibody availability, then this is not a major flaw.
2. In Fig 2A, a weak interaction is detected between HP1 and GR, however the authors conclude that there is no interaction. Is there a positive control that can be included to support this conclusion?
3. A bold conclusion from the manuscript is that the enzymatic activity of PRMT5 is not required for complex formation. This is very interesting, however as it stands, the data is not fully convincing. I am wondering if a longer incubation with GSK595 might alter findings because removal of methylation is thought to occur through protein turnover rather than a demethylase? More convincingly, what happens to GR/HP1 PLA in cells expressing methyl-deficient PRMT5? If PRMT5 activity is not required, is MEP50 recruited to the GR/HP1 /PRMT5 complex?
4. Can the genes listed in the GO presented in 4B be included as a table as this would be a useful resource to the community?
5. Chromatin immunoprecipitation of proteins that do not directly interact with DNA is challenging. Can additional controls for antibody specificity be included, such as knockdown of PRMT5 in PRMT5 ChIP samples?
6. The analysis presented in Figure 7A-C for GR is very interesting. Can the same analysis be carried out for PRMT5 and HP1 ?
7. Is there an ImageJ plug-in used quantification of PLA foci?

Reviewer #2 (Comments to the Authors (Required)):

In this manuscript, Nouredine et al. describe PRMT5 as a master regulator of GR function in TNBC cells. Surprisingly, although previously described that PRMT5 can methylate GR (Poulard et al. 2020), its role associated with HP1gamma recruitment and RNAPol II phosphorylation described here, turned out to be independent of its catalytic function. The paper describes an unknown mechanism of GR operation in TNBC whose rationale requires some additional experiments.

1-It is not clear to this reviewer that in Kwon et al 2010 it is shown that P-Ser 83-HP1gamma interacts with p-RNA-pol II, but rather total HP1gamma. Since HP1gamma phosphorylation is presented as key in the recruitment of RNAPol II, this point should be further clarified.

2-The authors concluded from Figure 2A that the GR-HP1gamma interaction is not direct, whereas, on the contrary, in panel F they claimed that PRMT5 interacts directly with GR. No such significant differences are observed in the blots to support these statements, appearing that in both cases the interaction is direct.

It's the opinion of this reviewer that a higher exposure of the blot in panel A will generate similar results to that of panel F. This point should be corrected or properly clarified.

In MDA-MB-231 cells, GR levels in Figure 2B are diametrically different compared to Figure 1D. This point should be clarified and properly justified.

3-In Figure 2D, the PRMT5-GR co-IP should be performed in the presence and absence of DEX to demonstrate whether or not the formation of this ternary complex is ligand-dependent. In turn, the presence of two bands in the GR blots should be clarified. On this regard, could this double band indicate that the GR in complex with HP1gamma and PRMT5 is post-translationally modified? This point should be addressed.

In addition, the molecular weight markers should be included

4-In Figure 4A, 275 GR-regulated genes are shown, does this set include up- and down genes? How do GR-dependent gene repression in TNBC cells fit within their model depicted in Figure 7E?

When PRMT5 and HP1gamma-dependent genes are shown (Figure 4A, venn diagrams). Are the genes that gain regulation upon knockdown included in these datasets? Can the authors identified genes where PRMT5 and HP1gamma have a repressive role in GR function? (see above) Are they involved in relevant cell/signaling pathways?

5-In Figure 4C, the basal levels of Serpine1, IGFBP3, PLAT and CCBE1 target genes are compromised in cells depleted of HP1gamma or PRMT5; how the authors justify this effect? Are PRMT5 and HP1gamma already loaded at target genes under uninduced conditions? Whether these effects are also detected at global level (RNA-seq) should be examined.

7-In the model (Figure 7E), how is the molecular mechanism behind HP1gamma recruitment/binding to PRMT5? Through specific methylation of the enzyme?

8-As P-Ser 83-HP1gamma serves as a marker for transcription elongation (Lomber et al 2006), how this fit with GR binding to regulated elements and recruitment of P-Pol II?

Reviewer #3 (Comments to the Authors (Required)):

In this paper, Nouredine et al. demonstrated that the GR/PRMT5/HP1 γ complex regulates the transcription of glucocorticoid target genes involved in cell migration that ultimately leads to the detrimental effects observed after complementary medication glucocorticoids treatment. This is a very neat work, with smooth logic flow and well-presented results. Without major caveat identified, the paper is recommended to publish in Life Science Alliance once the issues below being addressed by minor revision.

1, Fig. 1A, given the authors claimed that the indicated mechanism drives cell migration, inspection on metastasis-free survival will be more conceptually relevant in place of the "relapse-free" survival in general.

2, Fig. 2-3, the approach of PLA is heavily employed to visualize the co-localization of the proteins of interest, which might create a minor caveat. PLA is a start-of-art approach in highlighting protein-protein interactions. However, this approach had missed critical information such as the overall expression and distribution of the individual proteins. Regular IF staining will yet be helpful to reveal these information.

3, Fig. 4, the term "no fold change" and the underlying rationale to overlap comparison A, B and C was not clearly defined. It is confusing why the PRMT5/HP1 γ dep cells were compared with Dex-treated siNS cells rather than untreated siNS cells. Theoretically, if a "no fold change" is pursued, the phenotypes of dep cells are expected to be closer to untreated cells than the cells being treated by Dex.

4. Fig. 7C, the comparison between engrafts from metastases and primary tumors is a bit beyond the scope of this paper and in logic, not necessarily support the authors' hypothesis. The central hypothesis of this paper is that GR/PRMT5/HP1 γ complex may drive the migration of cancer cells, which may facilitate their escape from primary tumor. However, the hyperactivation of GR in distal metastases, even if truly observed, is not necessarily persisted for the mechanism being discussed (since the "migration capacity" is no longer needed once they have successfully colonized, whereby the "MET" concept raised recently), especially without evidence involving PRMT5 and HP1 γ .

Response to reviewers

We appreciate these constructive comments from the reviewers which have helped us to improve the manuscript.

Reviewer #1 (Comments to the Authors (Required)):

Triple negative breast cancer is currently treated using chemotherapy and immune modulators. Synthetic glucocorticoids (GCs) are given to patients to manage the side effects of these treatments. However, the use of GCs is now being questioned because of correlation with metastatic disease. In this paper, Nouredine et al., investigate the binding partners of glucocorticoid receptor (GR) in TNBC cell lines to understand how GCs mechanistically promote cell migration. They find that GR forms a complex with PRMT5 and HP1 γ , and that either component is required for GR-mediated chromatin localisation and gene expression, and cell migrations. Finally, using patient samples and tumours derived from breast PDX models, they very nicely demonstrate that the PRMT5-GR interaction is enhanced in chemotherapy treated rather than naive tumours. As such, the PRMT5-GR-HP1 γ complex could be prognosis for metastatic disease. This study addresses an important clinical question and could have significant translational impact. The experiments are well designed and the data of a very high standard and I very much enjoyed reading the manuscript. A minor criticism is the heavy reliance of PLA for protein interactions in cell lines, however the adaptation of this for FFPE tissue is an exciting development. I have a few comments:

1. I appreciate the controls put in place to determine that the PLA signal is not artefactual, however it would be good to see key cell line data supported by co-immunoprecipitation studies. The authors IP HP1 γ and detect PRMT5 and GR interactions (Fig 2D), can this be repeated with reciprocal co-IPs? If this is not possible due to antibody availability, then this is not a major flaw.

Response : We would first like to thank the referee for his/her kind comments. We primarily used PLA experiments that allow localization and quantification of the different interactions. However, to reply to the reviewer's request, we performed new co-immunoprecipitation experiments to support the PLA experiments. We first overexpressed GR, HP1 γ and PRMT5 in Cos-7 cells and assessed the interactions between the different proteins. The new Figure 2C shows that each protein is precipitated with the others, comforting the idea of a tripartite complex. To address the 3rd comment of the reviewer, we also performed an IP with a MEP50 antibody and were able to precipitate GR, HP1 γ and PRMT5. In addition, as requested by the reviewer 2, we also performed the experiment in endogenous conditions on MDA-MB-231 cell lines in the presence or absence of Dex. In the new Figure 2F, an IP was performed with a

PRMT5 antibody and demonstrated that PRMT5 interacts with (i) HP1 γ upon Dex treatment and (ii) GR independently of Dex treatment. These results are in accordance with the other results presented in the paper as we saw that (i) PRMT5/HP1 γ interaction increases after Dex treatment by PLA (Fig 2B) and (ii) that the localization of the interaction GR/PRMT5 changes upon Dex treatment in PLA (Fig 2A) and is not affected by Dex treatment in GST pull-down (Fig 2D).

2. In Fig 2A, a weak interaction is detected between HP1 γ and GR, however the authors conclude that there is no interaction. Is there a positive control that can be included to support this conclusion?

Response : *We agree that this result was not striking and we decided to place the previous Fig 2A in the supplemental figure EV4A. In order to support our conclusions, we divided GR into fragments following the main steroid receptor domains (Figure EV4B) and performed GST pull-down experiments between this GST-GR domain and Grip1, a well-known GR coactivator as a positive control (Figure EV4D) and HP1 γ (Figure EV4C). As expected, Grip1 interacts with GR1. However, HP1 γ does not interact with any of the GR domains. In addition, using these domains, we were able to demonstrate that PRMT5 interacts mainly with the GR1 domain (Figure EV4F).*

3. A bold conclusion from the manuscript is that the enzymatic activity of PRMT5 is not required for complex formation. This is very interesting, however as it stands, the data is not fully convincing. I am wondering if a longer incubation with GSK595 might alter findings because removal of methylation is thought to occur through protein turnover rather than a demethylase? More convincingly, what happens to GR/HP1 γ PLA in cells expressing methyl-deficient PRMT5? If PRMT5 activity is not required, is MEP50 recruited to the GR/HP1 γ /PRMT5 complex?

Response : *As suggested by the reviewer, we performed an experiment with 6 days of exposure with GSK595 and found that although it strongly decreased the global symmetric demethylation level in cells, GR/HP1 γ interaction was not affected. In addition, it does not affect protein expression. We decided to remove the previous experiment done with 3 days of GSK595 and replace it by 6 days of GSK595 (Figure 3D).*

We did not choose to perform experiments with a mutant PRMT5 as we preferred to maintain endogenous conditions.

As discussed earlier, we also performed an IP with a MEP50 antibody and were able to precipitate GR, HP1 γ and PRMT5.

4. Can the genes listed in the GO presented in 4B be included as a table as this would be a useful resource to the community?

Response : As suggested by the reviewer we have now including the data in the supplemental Table 1.

5. Chromatin immunoprecipitation of proteins that do not directly interact with DNA is challenging. Can additional controls for antibody specificity be included, such as knockdown of PRMT5 in PRMT5 ChIP samples?

Response : Chromatin immunoprecipitation of coregulators is indeed challenging. We added a control as requested by the reviewer: PRMT5 ChIP on samples where PRMT5 was knockdown or not. The result has been inserted into the supplemental Figure EV6C.

6. The analysis presented in Figure 7A-C for GR is very interesting. Can the same analysis be carried out for PRMT5 and HP1 γ ?

Response : Based on the reviewer's request, we analyzed HP1 γ and PRMT5 expression in our PDX models. The results were added in Figure 7A-7B-8EVB-8EVC. PRMT5 and HP1 γ expression remained the same between metastasis and primary tumors. There was a higher expression in luminal tumors than in TNBC. Interestingly as for GR, the expression of HP1 γ was significantly higher in metaplastic tumors compared to other subtypes. In addition, PRMT5 expression was higher in apocrine tumors compared to other subtypes.

7. Is there an ImageJ plug-in used quantification of PLA foci?

Response : Yes, the PLA foci were quantified using ImageJ. The analytical process was detailed in Poulard et al, Methods, 2020. We emphasized this point in the manuscript in the method section.

Reviewer #2 (Comments to the Authors (Required)):

In this manuscript, Nouredine et al. describe PRMT5 as a master regulator of GR function in TNBC cells. Surprisingly, although previously described that PRMT5 can methylate GR (Poulard et al. 2020), its role associated with HP1 γ recruitment and RNAPol II phosphorylation described here, turned out to be independent of its catalytic function. The paper describes an unknown mechanism of GR operation in TNBC whose rationale requires some additional experiments.

1-It is not clear to this reviewer that in Kwon et al 2010 it is shown that P-Ser 83-HP1 γ

interacts with p-RNA-pol II, but rather total HP1gamma. Since HP1gamma phosphorylation is presented as key in the recruitment of RNAPol II, this point should be further clarified.

Response : *We thank the reviewer for his/her comment, we apologize for the error made in this reference. Indeed, the correct reference is Lomber et al, Nat Cell Biol, 2006. In this paper, the authors demonstrated by colP that p-S83-HP1 γ interacts with p-S5-RNAPolIII.*

2-The authors concluded from Figure 2A that the GR-HP1gamma interaction is not direct, whereas, on the contrary, in panel F they claimed that PRMT5 interacts directly with GR. No such significant differences are observed in the blots to support these statements, appearing that in both cases the interaction is direct.

It's the opinion of this reviewer that a higher exposure of the blot in panel A will generate similar results to that of panel F. This point should be corrected or properly clarified.

In MDA-MB-231 cells, GR levels in Figure 2B are diametrically different compared to Figure 1D. This point should be clarified and properly justified.

Response : *We agree that this result was not striking and we decided to place the previous Fig 2A in the supplemental figure EV4A. In order to support our conclusions, we divided GR into fragments following the main steroid receptor domains (Figure EV4B) and performed GST pull-down experiments between this GST-GR domain and Grip1, a well-known GR coactivator as a positive control (Figure EV4D) and HP1 γ (Figure EV4C). As expected, Grip1 interacts with GR1. However, HP1 γ does not interact with any of the GR domains. In addition, using these domains, we were able to demonstrate that PRMT5 interacts mainly with the GR1 domain (Figure EV4F).*

The two figures the reviewer pointed out are indeed different in the sense that there is a strong difference in the exposure time. Indeed, in Figure 1D, we overexposed the GR panel in order to see if we detect a little bit of GR in BT20 and HCC-1937. The point was to show that even if there is a small level of GR, the interaction occurs. Original data with different exposure times should clarify this (Original data file).

3-In Figure 2D, the PRMT5-GR co-IP should be performed in the presence and absence of DEX to demonstrate whether or not the formation of this ternary complex is ligand-dependent. In turn, the presence of two bands in the GR blots should be clarified. On this regard, could this double band indicate that the GR in complex with HP1gamma and PRMT5 is post-translationally modified? This point should be addressed.

In addition, the molecular weight markers should be included

Response

We have added extra experiments to provide a better answer to this question. We performed the experiment in endogenous conditions on MDA-MB-231 cell lines in the presence or

absence of Dex. In the new Figure 2F, an IP was performed with a PRMT5 antibody and demonstrated that PRMT5 interacts with (i) HP1 γ upon Dex treatment and (ii) GR independently of Dex treatment. These results are in accordance with the other results presented in the paper as we saw that (i) PRMT5/HP1 γ interaction increases after Dex treatment by PLA (Fig 2B) and (ii) that the localization of the interaction GR/PRMT5 changes upon Dex treatment in PLA (Fig 2A) and is not affected by Dex treatment in GST pull-down (Fig 2D).

Thanks to the reviewer's comment, we realized that the molecular weight markers were missing and could impede the interpretation of the colP. For this reason, we included the molecular weight markers in all the figures. Also, the molecular weight would have helped the reviewer to see that there is a difference in the gel migration. Indeed, in order to have a better separation between PRMT5 (72kDa) and GR (95kDa) in this colP experiment, we used 6% polyacrylamide gel that separated relatively well the two isoforms of GR. Using a 10 or 12% gel, the two bands are quite close as in figure 3A or 3C, with upper band generally more intense, mainly express in low exposure time. In some cases, we used premade gel (4-20% polyacrylamide gels) that does not allow a good separation between the two isoforms as is the case in figure 2B. In addition, the experiments were performed using two different antibodies (GR G-5 (Santa Cruz sc-393232), GR (Cell signaling #12041). GR from cell signaling recognize only the upper band corresponding to the main isoform used for example for Fig 1F, 2A, 2B.

4-In Figure 4A, 275 GR-regulated genes are shown, does this set include up- and down genes? How do GR-dependent gene repression in TNBC cells fit within their model depicted in Figure 7E? When PRMT5 and HP1 γ -dependent genes are shown (Figure 4A, venn diagrams). Are the genes that gain regulation upon knockdown included in these datasets? Can the authors identified genes where PRMT5 and HP1 γ have a repressive role in GR function? (see above) Are they involved in relevant cell/signaling pathways?

Response : Thanks to this comment we realized that the legend of Figure 4A and manuscript were not sufficiently detailed. We added more information in the legend and in the text. Of the 275 GR-regulated genes that were up- and down-regulated, genes were included to globally determine the impact of PRMT5 and HP1 γ depletion. Of the 89 genes analyzed for pathway analysis, all up- and down-regulated genes were included to determine the impact of PRMT5 and HP1 γ depletion on the Dex-regulated phenotype. Of course, the paper focused on the coactivated activity of HP1 γ and PRMT5. However, PRMT5 and HP1 γ corepressive activity could participate in the phenotype. For instance, BTG1 overexpression was demonstrated to inhibit cell migration and invasion in endometrial and breast cancer (Li Y, Cancer cell int, 2020-Sheng SH, Tumor Biol, 2014). In addition, a low expression of BTG1 in endometrial and breast cancer is associated with a poorer prognosis. In our dataset, this gene is down-regulated by Dex induction. As PRMT5 and HP1 γ depletion prevents this depletion, it suggests that PRMT5 and HP1 γ work in this case as corepressors, their well-known function (for review

Schoelz J, *Epigenetics & chromatin*, 2022-Motolani, Life, 2021). As we did not validate this aspect in our work, we did not change the model presented in Fig7D.

5-In Figure 4C, the basal levels of Serpine1, IGFBP3, PLAT and CCBE1 target genes are compromised in cells depleted of HP1 γ or PRMT5; how the authors justify this effect? Are PRMT5 and HP1 γ already loaded at target genes under uninduced conditions? Whether these effects are also detected at global level (RNA-seq) should be examined.

Response : As previously reported (Poulard et al, PNAS, 2019-Cell death dis 2018-Bittencourt 2012), coregulator depletion can significantly change mRNA levels of some genes in the absence of Dex. For this reason, we analyzed Dex-induced cell phenotypes according to the levels of gene products after Dex treatment rather than the Dex-induced fold-change. Indeed, the ultimate expression level of a gene upon addition of Dex governs the biological response. However, from a mechanistic point of view, we believe that PRMT5 and HP1 γ could already be loaded under control conditions. For HP1 γ , using siRNA approach, we previously demonstrated that HP1 γ ChIP signal without Dex induction could be decreased indicating constitutive HP1 γ occupancy on chromatin (Poulard et al, EMBO rep, 2007) and this depletion was associated with a decrease in the basal level of the target genes. Using siRNA against PRMT5, we saw a similar pattern for PRMT5 recruitment under control conditions for the PLAT gene (new supplemental figure EV6C).

7-In the model (Figure 7E), how is the molecular mechanism behind HP1 γ recruitment/binding to PRMT5? Through specific methylation of the enzyme?

Response : We currently have no information to explain how HP1 γ is recruited by PRMT5. We cannot rule out the fact that PRMT5 could be modified by lysine methyltransferase serving as a platform for HP1 γ recruitment. Indeed, HP1 γ is a protein that binds methylated residues through their chromodomain, as G9ame2 or H3K9me2 (Poulard et al, EMBO Rep, 2017-Sampath, Mol Cell 2007-Lachner, Nature, 2001). Additional experiments would be necessary to address this question fully.

8-As P-Ser 83-HP1 γ serves as a marker for transcription elongation (Lomberk et al 2006), how this fit with GR binding to regulated elements and recruitment of P-Pol II?

Response : Through a comment made by the first reviewer, we realized that we had made an error in this reference. Indeed, Lomberk et al demonstrated by coIP that p-S83-HP1 γ interacts with p-S5-RNAPolIII. As we discussed in the manuscript, our results (Figure 5D) in addition to previously published data (Poulard EMBO Rep 2017-Schoelz J, *Epigenetics & Chromatin*,

2022), allowed us to build the model in which p-S83-HP1 γ allows the recruitment of p-S5-RNAPolIII.

Reviewer #3 (Comments to the Authors (Required)):

In this paper, Nouredine et al. demonstrated that the GR/PRMT5/HP1 γ complex regulates the transcription of glucocorticoid target genes involved in cell migration that ultimately leads to the detrimental effects observed after complementary medication glucocorticoids treatment. This is a very neat work, with smooth logic flow and well-presented results. Without major caveat identified, the paper is recommended to publish in Life Science Alliance once the issues below being addressed by minor revision.

1, Fig. 1A, given the authors claimed that the indicated mechanism drives cell migration, inspection on metastasis-free survival will be more conceptually relevant in place of the "relapse-free" survival in general.

Response : We would like to thank the reviewer for his/her positive feedback and comment, we analyzed the DMFS (distant metastasis free survival). However, we did not see anything relevant, probably due to the small number of patients available.

2, Fig. 2-3, the approach of PLA is heavily employed to visualize the co-localization of the proteins of interest, which might create a minor caveat. PLA is a start-of-art approach in highlighting protein-protein interactions. However, this approach had missed critical information such as the overall expression and distribution of the individual proteins. Regular IF staining will yet be helpful to reveal these informations.

Response : The PLA approach was chosen as it allows localization and quantification of protein-protein interactions. However, we agree with the reviewer that it does not provide information about the overall expression and distribution of the individual proteins. To circumvent this issue, we added regular IF staining in supplemental figure EV5. This IF showed that GR is mainly localized in the cytoplasm without ligand and translocates to the nucleus upon Dex treatment, which is concordant with the literature. However, HP1 γ remains in the nucleus of cells and PRMT5 is mainly localized in the cytoplasm, with a small pool of protein in the nucleus and this dual localization does not change upon Dex treatment. Additionally, we analyzed if the inhibition of PRMT5 changes the localization of GR and HP1 γ as PRMT5 was shown to be a scaffolding protein in this complex and found no changes.

3, Fig. 4, the term "no fold change" and the underlying rationale to overlap comparison A, B and C was not clearly defined. It is confusing why the PRMT5/HP1 γ dep cells were compared with Dex-treated siNS cells rather than untreated siNS cells. Theoretically, if a "no fold change" is pursued, the phenotypes of dep cells are expected to be closer to untreated cells than the cells being treated by Dex.

Response : The statement "No fold change" was incorrectly used. Indeed, the green Venn diagram represents the number of HP1 γ -regulated genes with significantly different expression (q -value ≤ 0.05 , no fold change cut-off was imposed) in Dex-treated cells expressing siHP1 γ versus Dex-treated cells expressing siNS. The blue Venn diagram represents the number of PRMT5-regulated genes with significantly different expression (q -value ≤ 0.05 , no fold change cut-off was imposed) in Dex-treated cells expressing siPRMT5 versus Dex-treated cells expressing siNS. We have now clarified this in the legend and in the manuscript.

4. Fig. 7C, the comparison between engrafts from metastases and primary tumors is a bit beyond the scope of this paper and in logic, not necessarily support the authors' hypothesis. The central hypothesis of this paper is that GR/PRMT5/HP1 γ complex may drive the migration of cancer cells, which may facilitate their escape from primary tumor. However, the hyperactivation of GR in distal metastases, even if truly observed, is not necessarily persisted for the mechanism being discussed (since the "migration capacity" is no longer needed once they have successfully colonized, whereby the "MET" concept raised recently), especially without evidence involving PRMT5 and HP1 γ .

Response : We agree with the reviewer but decided to keep it in the paper as the other reviewers did not object to it and these elements are interesting for the scientific community.

July 14, 2023

RE: Life Science Alliance Manuscript #LSA-2023-02009-TR

Dr. Coralie Poulard
Cancer Center of Lyon
Inserm U1052 CNRS UMR 5286
Centre Léon Bérard, 28 rue Laennec
Lyon 69373 Lyon Cedex 08
France

Dear Dr. Poulard,

Thank you for submitting your revised manuscript entitled "PRMT5 triggers glucocorticoid-induced cell migration in Triple Negative Breast Cancer". We would be happy to publish your paper in Life Science Alliance pending final revisions necessary to meet our formatting guidelines.

- please upload all figure files as individual ones, including the supplementary figure files; all figure legends should only appear in the main manuscript file
- please add your main, supplementary figure, and table legends to the main manuscript text after the references section
- please add ORCID ID for the corresponding (and secondary corresponding) author--you should have received instructions on how to do so
- please add a Summary Blurb/Alternate Abstract to our system
- please add the Twitter handle of your host institute/organization as well as your own or/and one of the authors in our system
- there is a discrepancy in the name presentation of one co-author; please correct (Ha Thuy Pham in the manuscript file vs. Thuy Ha Pham in the system)
- LSA allows supplementary figures, but no EV Figures; please update your callouts for the Supplementary Figures in the manuscript Fig EV1A=Fig S1A; while supplementary figures use the system supplementary Fig S1;
- please add callouts for Figures S3A-D; S5A-B to your main manuscript text
- please add a Data Availability Statement at the end of the Materials and Methods to indicate the GEO accession information for the RNA-seq data
- please remove the entire "The paper explained" section

A. FINAL FILES:

B. MANUSCRIPT ORGANIZATION AND FORMATTING:

Sincerely,

Reviewer #1 (Comments to the Authors (Required)):

The authors have addressed all the reviewers concerns and the manuscript is now suitable for publication.

Reviewer #2 (Comments to the Authors (Required)):

In the revised version of the article, the authors have successfully corrected the erroneous reference, provided a more comprehensive elucidation of the GR interaction with both HP1g and PRMT5, incorporated additional data regarding the ligand-induced activation of GR in these interactions, and offered valuable insights into the proposed mechanism's impact on the 275 genes dependent on GR. These newly obtained results provide robust support for the proposed hypotheses. Consequently, the authors have addressed all of my concerns, and I am pleased to endorse the revised manuscript for publication in Life Science Alliance.

Reviewer #3 (Comments to the Authors (Required)):

The revision has addressed my prior concerns.

July 24, 2023

RE: Life Science Alliance Manuscript #LSA-2023-02009-TRR

Dr. Coralie Poulard
Cancer Center of Lyon
Inserm U1052 CNRS UMR 5286
Centre Léon Bérard, 28 rue Laennec
Lyon 69373 Lyon Cedex 08
France

Dear Dr. Poulard,

Thank you for submitting your Research Article entitled "PRMT5 triggers glucocorticoid-induced cell migration in Triple Negative Breast Cancer". It is a pleasure to let you know that your manuscript is now accepted for publication in Life Science Alliance. Congratulations on this interesting work.

DISTRIBUTION OF MATERIALS:

Again, congratulations on a very nice paper. I hope you found the review process to be constructive and are pleased with how the manuscript was handled editorially. We look forward to future exciting submissions from your lab.

Sincerely,
